# QUANTIFYING MECHANISTIC GAPS IN ALGORITHMIC REASONING VIA NEURAL COMPILATION

## ABSTRACT

Neural networks can achieve high prediction accuracy on algorithmic reasoning tasks, yet even effective models fail to faithfully replicate ground-truth mechanisms, despite the fact that the training data contains adequate information to learn the underlying algorithms faithfully. We refer to this as the *mechanistic gap*, which we analyze by introducing neural compilation for GNNs, which is a novel technique that analytically encodes source algorithms into network parameters, enabling exact computation and direct comparison with conventionally trained models. Specifically, we analyze graph attention networks (GATv2), because of their high performance on algorithmic reasoning, mathematical similarity to the transformer architecture, and established use in augmenting transformers for NAR. Our analysis selects algorithms from the CLRS algorithmic reasoning benchmark: BFS, DFS, and Bellman-Ford, which span effective and algorithmically aligned algorithms. We quantify faithfulness in two ways: external trace predictions, and internal attention mechanism similarity. We demonstrate that there are mechanistic gaps even for algorithmically-aligned parallel algorithms like BFS, which achieve near-perfect accuracy but deviate internally from compiled versions.

## 1 INTRODUCTION

Mechanistic faithfulness guarantees generalization and robustness, and better understanding it is critical in building artificial intelligence that can reason. We study this in the realm of Neural Algorithmic Reasoning (NAR), which studies the ability of neural networks to learn algorithmic reasoning tasks. The main purpose of this paper is in measuring mechanistic faithfulness on these algorithmic tasks: many models can learn effective approximations to these algorithms, but do they actually learn the intended behavior? This is mechanistic faithfulness. For example, trained GATv2 predicts the Bellman-Ford shortest paths algorithm with 87% accuracy, but does it actually learn the dynamic programming mechanism correctly? How can we quantify these mechanistic gaps?

We answer this in two ways: first, we analyze external trace predictions, and second, we use neural compilation to compare learned algorithms to a ground truth, which allows us to quantify internal mechanistic similarity. First, the CLRS benchmark [1] includes algorithmic traces (also called hints). These traces describe the intermediate states and operations of an algorithm, such as the partially explored graph for breadth-first search (BFS).In principle, supervised training on traces enables learning a mechanistically correct solution, at least in the sense that the model is given adequate information to reproduce the target algorithm. In practice, presenting this data does not explicitly induce reasoning, and in some cases models perform better without it [1, 2]. Trace predictions can measure faithfulness, but only externally. Accordingly, we quantify internal faithfulness through similarity of the GATv2 attention mechanism to a neurally-compiled ground truth.

**Neural Compilation** The upper-bound expressivity of many neural network architectures is established, but expressivity does not guarantee that gradient-based optimization will find either effective or faithful algorithms [3]. The focus of this paper is in understanding this gap by using neural compilation as an analysis tool. Neural compilation is a technique for converting programs into neural network parameters that compute the original program [4, 5, 6, 7, 8, 9]. Neurally compiled programs are implicit expressivity proofs, ground-truth references, and optima of the underlying optimization problem [9]. We use neural compilation to better understand the mechanistic gap by analyzing intermediate behaviors, primarily the attention mechanism in GATv2.

**Defining Mechanistic Faithfulness: Unique Solutions in Algorithmic Phase Space**    Neural compilation allows us to be more precise in defining mechanistic faithfulness, because it gives us ground truth to compare against. We draw upon the idea of *algorithmic phase space* ([10]): neural network parameters describe a low-level program space that admits a vast diversity of solutions. Neural compilation allows specifying abstract program behaviors, and the set of low-level parameters which produce them. In particular, our analysis focuses on the attention mechanism in GATv2, which is effectively a (semi) interpretable symbolic layer. Having a neurally compiled ground truth enables quantifying the mechanistic gap directly by comparing attention activations (Equation 24).

**Does Algorithmic Alignment Confer Faithfulness?**    A major factor explaining expressivity-trainability gaps is *algorithmic alignment*, the idea that certain neural networks are more efficient at learning particular algorithms [11]. For example, graph neural networks, especially GATv2, are particularly suited for graph-based dynamic programming tasks [12, 13]. Furthermore, in general it is easier to learn parallel algorithms than it is to learn inherently sequential ones, especially for GNNs. We refer to this as NC-Learnability [14]. However, algorithmic alignment is formulated in terms of a sample-complexity bound for *accuracy*, not faithfulness explicitly. Even though architectural similarity seems like it might confer mechanistic faithfulness [3], our analysis finds that there are still mechanistic gaps even under algorithmic alignment and parallelism.

## 1.1 CONTRIBUTIONS

1. A neural compilation technique for GATv2, demonstrated on BFS and Bellman-Ford.

2. Metrics for quantifying mechanistic gaps: external trace prediction accuracy and internal attention mechanism similarity.

3. Empirical evidence showing no correlation between prediction accuracy and faithfulness, even for aligned parallel algorithms like BFS.

## 2 RELATED WORK

**Differentiable Computing**    Previous work has considered differentiable models of computation, such as LSTMS or other RNNs [15]. This was expanded by Neural Turing Machines and Hybrid Differentiable Computers [16, 17]. However, sequential models of computation are often exceptionally difficult to train, which was a big factor in the invention of transformers [18, 19, 20, 3].

**Neural Algorithmic Reasoning**    Neural algorithmic reasoning (NAR) has evolved through benchmarks and techniques that enhance model alignment with algorithmic tasks, particularly on graph structures. While early GNNs were focused on modeling structured data, later variants were inspired by differentiable computing, but in practice can be far more effective than their original counterparts. Originally, GNNs were proposed in [21]. However, they have seen a rich variety of extensions [22]. Notably, Deep Sets introduced permutation invariance [23], Message-Passing Neural Networks (MPNN) introduced a framework for various models of graph computation [24], which Triplet MPNN extended with several architecture modifications, such as gating, triplet reasoning, and problem specific decoders [25]. Separately, GAT introduced a self-attention mechanism [12]. GATv2 generalized this to dynamic attention [13]. Finally, Pointer Graph networks enabled processing graphs with dynamic topology [26]. Together, the CLRS benchmark captures many of these improvements, and provides these models as baselines.

These models have high generalization on neural algorithmic reasoning tasks [1, 13]. This is critical, as it makes our comparisons meaningful. We select GATv2 because of it has relatively high performance, and the attention mechanism is mathematically similar to the attention mechanism in a transformer, the primary difference being that graph adjacency is used to mask attention coefficients for GATv2, while standard transformers assume a fully connected topology.

CLRS gives us several interesting cases to study: BFS, where trained performance is nearly perfect; Bellman-Ford, where trained performance is high, but not perfect, and DFS, where trained performance struggles significantly. BFS in particular is the most interesting, because the near-perfect learned algorithm does not faithfully learn the underlying algorithmic mechanism, even though BFS is relatively simple, algorithmically aligned with GATv2, and proven to be in NC [27, 11].

**Neural Compilation**   Neural Compilation is a technique for transforming conventional computer programs into neural network parameters that compute the input algorithm. Fundamentally, neural compilation constructs an injective function (compiler) that maps program space to parameter space: $\mathcal{C} : \Gamma \mapsto \Theta$ so that the behaviors of a program $\gamma \in \Gamma$ and $f(\theta), \theta \in \Theta$ are consistent on all inputs (where $f$ is a neural network architecture, $\theta$ its parameters, and $\Theta$ the parameter space, e.g. $\mathbb{R}^p$). The earliest results in neural compilation stem from [4] and [5]. Decades later, [6] developed *adaptive* neural compilation, for initializing networks with compiled solutions and then further training them. After the invention of the transformer architecture, there became significant interest in characterizing its internal mechanisms through programs, e.g. *mechanistic interpretability*. From this came RASP, TRACR, and ALTA [7, 8, 9], which compile a domain-specific language into transformer parameters. Notably, ALTA ([9]) includes comparisons between learned and compiled algorithms, and [28] includes theoretical graph-algorithm results.

**Expressivity and Trainability**   Many papers establish theoretical upper bounds of neural network expressivity [29, 30, 31, 28], dating back to the origins of the field [32, 4]. However, it is more difficult to make substantive statements about trainability. In practice, theoretical expressivity bounds are not reached for a wide variety of models [3]. For example, [30] establishes that transformers can express $\texttt{TC}^0$, but [9] shows that they struggle to learn length-general parity from data. Within neural algorithmic reasoning, [14] and [11] support GNNs potentially expressing algorithms in $\texttt{PRAM}$ ($\texttt{NC}$), but this has not been formally proven. Beyond learning effective solutions that saturate expressivity bounds, we also wish to learn mechanistically faithful algorithms. Mechanistic faithfulness implies generalization and saturation of expressivity.

**Critical Work on Neural Network Reasoning**   Given the high-profile nature of neural networks, especially language models, several papers criticize their reasoning ability in the hope of understanding how to improve them [33, 34, 35, 36]. This motivates mechanistic interpretability studies and future work, but also grounds expectations about the capabilities of these systems. Similarly, the quantitative measures of mechanistic faithfulness we introduce are intended to play a role in improving algorithmic reasoning.

**Mechanistic Interpretability**   While neural compilation techniques have their roots in differentiable computing, their application to mechanistic interpretability is a more recent phenomenon, inspired several other approaches for interpreting neural network behavior, especially that of large language models. Fundamentally, mechanistic interpretability aims to reverse-engineer learned behavior into an interpretable form. In the most general case, this behavior would be described as abstract computer programs (e.g. neural *decompilation*). However, this is fundamentally difficult, given that neural network computation tends to be dense, parallel, and polysemantic. Some work characterizes "circuits", e.g. sub-paths of a neural network that correspond to a particular behavior [37, 38, 39]. Other techniques try to extract categorical variables from dense, polysemantic representations [40]. Notably [10] attempts to categorize the algorithmic phase space (solution space) of addition algorithms, similar to ALTA's analysis of learned parity functions [9]. Work on "grokking" attempts to capture phase-shifts in neural network generalization, e.g. where a faithful version of an algorithm gradually replaces memorized data [41, 39, 10, 42].

For neural algorithmic reasoning specifically, [43] introduces the concept of the *scalar bottleneck*, a potential explanation for why faithful algorithms are difficult to learn, which is later refined by [44], which proposes learning algorithm ensembles. The scalar bottleneck hypothesis, as well as the idea of algorithmic phase space, help explain why learned models favor dense representations over sparse, faithful ones, complementing our empirical evidence from compiled comparisons.

## 3   METHODS

Our methods section uses Einstein notation with dimension annotations. For example:

$$\underset{\text{name}}{\overset{m \times n}{A_{ik}}} = \underset{\text{name}}{\overset{m \times l}{B_{ij}}} \; \underset{\text{name}}{\overset{l \times n}{C_{jk}}} \tag{1}$$

Depicts a matrix multiplication by implying summation of the dimension $j$ (size $l$). While this is quite verbose, it ensures clarity when describing higher-dimensional tensor contractions or complex operations. See [45] for an accessible reference.

## 3.1 Background: Graph Attention Networks (GATv2)

For a graph $G = (\mathcal{V}, \mathcal{E})$ with $n$ vertices, graph attention networks work by iteratively refining vector representations $h$ at each vertex (collectively, $H$) by exchanging information between vertices according to the graph topology and a learned attention mechanism [12, 13]. The model receives input $\mathcal{V}$ of dimension $n \times f_{\mathcal{V}}$, representing a vector of size $f_{\mathcal{V}}$ for each vertex, and edge information $\mathcal{E}$, which is an $n \times n \times f_{\mathcal{E}}$ tensor, which similarly has a feature vector for each edge. We define the graph topology with an adjacency matrix $A$, which is an $n \times n$ matrix with binary entries. Also, the graph contains metadata in a vector $g$, with dimension $f_g$. While feature dimensions can vary, we use $f$ where they can be implied by context. Consider a graph attention network with hidden size $s$ and $d$ attention heads. For convenience, let $m = \frac{s}{d}$ (the number of attention heads, $d$, must divide $s$). For this network, the parameters $\theta$ are:

$$\theta = \begin{pmatrix} \overset{s\times(f+s)}{\underset{\text{val}}{W}} & \overset{s\times(f+s)}{\underset{\text{in}}{W}} & \overset{s\times(f+s)}{\underset{\text{out}}{W}} & \overset{s\times f}{\underset{\text{edge}}{W}} & \overset{s\times f}{\underset{\text{meta}}{W}} & \overset{s\times(f+s)}{\underset{\text{skip}}{W}} & \overset{d\times m}{\underset{\text{attn}}{\omega}} \end{pmatrix} \tag{2}$$

GATv2 relies on an attention mechanism which selects information to pass between adjacent vertices. This is calculated as a function of parameters, features, the adjacency matrix, and hidden state:

$$\overset{n\times n\times d}{\underset{\text{attn}}{\alpha}} = \mathcal{F}(\theta, \mathcal{V}, \mathcal{E}, g, A, H) \tag{3}$$

First, the model computes intermediate values $\nu$, which are candidates for new hidden representations.

$$\overset{n\times f}{\underset{\text{input}}{\mathcal{V}}} \quad \overset{n\times s}{\underset{\text{hidden}}{H}} \quad \overset{n\times(f+s)}{\underset{\text{concat}}{C}} = [\mathcal{V}|H] \qquad \overset{n\times s}{\nu_{il}} = \underset{\text{val}}{W_{lk}} C_{ik} \tag{4}$$

Second, the model computes two intermediate representations from the concatenated node features. These represent incoming and outgoing information to and from each node. Then, the model computes separate intermediate representations for edges and graph metadata:

$$\overset{n\times s}{\underset{\text{in}}{z_{ip}}} = \underset{\text{in}}{W_{pk}} C_{ik} \qquad \overset{n\times s}{\underset{\text{out}}{z_{ip}}} = \underset{\text{out}}{W_{pk}} C_{ik} \qquad \overset{n\times n\times s}{\underset{\text{edge}}{z_{ijp}}} = \underset{\text{edge}}{W_{ph}} \mathcal{E}_{ijh} \qquad \overset{s}{\underset{\text{meta}}{z_q}} = \underset{\text{meta}}{W_{qr}} g_r \tag{5}$$

These intermediate representations are combined into a single tensor, $\zeta$, using broadcasting.

$$\overset{n\times n\times s}{\underset{\text{pre attn}}{\zeta}} = \overset{1\times n\times s}{\underset{\text{in}}{z}} + \overset{n\times 1\times s}{\underset{\text{out}}{z}} + \overset{n\times n\times s}{\underset{\text{edge}}{z}} + \overset{1\times 1\times s}{\underset{\text{meta}}{z}} \tag{6}$$

Then, $\zeta$ is used to compute unnormalized attention scores, $a$, using the attention heads $\omega$. First $\zeta$ is split into the tensor $n \times n \times d \times m$, to provide a vector of size $m$ to each head:

$$\overset{n\times n\times d}{a_{ijh}} = \omega_{ho} \sigma(\zeta)_{ijho} \tag{7}$$

Where $\sigma$ is a leaky ReLU activation [46]. To enforce graph topology, we create a bias tensor from the adjacency matrix:

$$\overset{n\times n}{\underset{\text{bias}}{\beta}} = c * (A - 1) \tag{8}$$

where $c$ is a large constant, e.x. 1e9. This is used to nullify attention scores between unconnected nodes. Then, the final scores are normalized with softmax ($\beta$ is broadcast in the final dimension as a $n \times n \times 1$, e.g. for each attention head):

$$\overset{n\times n\times d}{\underset{\text{attn}}{\alpha}} = \texttt{softmax}(a + \beta) \tag{9}$$

Finally, these attention scores are used to select values from the candidates computes earlier. Selected values from different heads are summed together, and a skip connection propagates other information into the next hidden representation. Note that $\underset{\text{val}}{\nu}$ is reshaped into a $n \times d \times m$ tensor for the $d$ attention heads, and then $\underset{\text{select}}{\nu}$ is reshaped back into a $(n \times s)$ tensor to match $\underset{\text{skip}}{\nu}$.

$$\overset{n\times d\times m}{\underset{\text{select}}{\nu_{jho}}} = \alpha_{ijh} \overset{n\times d\times m}{\underset{\text{val}}{\nu_{iho}}} \qquad \overset{n\times s}{\underset{\text{skip}}{\nu_{il}}} = \underset{\text{skip}}{W_{lk}} C_{ik} \qquad \underset{\text{next}}{H} = \overset{n\times s}{\underset{\text{select}}{\nu}} + \sigma\left(\underset{\text{skip}}{\nu}\right) \tag{10}$$

Finally, the new $H$ is normalized with layer norm, completing a single iteration of graph attention.

## 3.2 Architecture Modifications

Neural compilation revealed certain aspects of the GATv2 architecture which can affect the ability to express particular algorithms naturally. These modifications reflect previous findings in NAR [47, 25]. Most notably, it was clear that candidate values $\nu$ for graph attention (Equation 4) are not a function of the edge features, $\mathcal{E}$, meaning there is not a natural way to store or process edge information in the hidden states of the model, outside of the attention mechanism. However, using edge information makes it significantly easier to compute cumulative edge distances when running algorithms like Bellman-Ford. Specifically we introduce a linear layer $\underset{\text{info}}{W}$ which operates on edge features:

$$\underset{\text{mid}}{\overset{n \times n \times m}{\mathcal{E}_{ijk}}} = \underset{\text{info}}{\overset{m \times f}{W_{lk}}} \underset{\text{input}}{\overset{n \times n \times f}{\mathcal{E}_{ijl}}} \tag{11}$$

$$\underset{\text{edge}}{\overset{n \times d \times m}{\nu_{ihk}}} = \left( \alpha \odot \underset{\text{mid}}{\overset{n \times n \times d \times m}{\mathcal{E}}} \right)_{ijhk} \tag{12}$$

$$\underset{\text{final}}{H} = \underset{\text{next}}{H} + \underset{\text{edge}}{\overset{n \times s}{\nu}} \tag{13}$$

In these equations, $\underset{\text{info}}{W}$ encodes edge information to include in each node representation, the attention coefficients $\alpha$ select it (the hadamard product, $\odot$ broadcasts in the head dimension, $d$), and then the incoming edge dimension is summed to match the dimensions of the hidden states.

We also experiment with adding a pre-attention bias $\mathcal{B}$ (dimension $n \times n$), which has similar behavior to the bias matrix $\beta$ calculated from the adjacency matrix in Equation 8, except that it is learned:

$$\underset{\text{post}}{\zeta} = \underset{\text{pre}}{\zeta} + \mathcal{B} \tag{14}$$

Introducing $\mathcal{B}$ allows algorithms to have more consistent default behavior, for instance nodes that are not currently being explored are expected to remain unchanged, and adding a bias layer before the attention weights makes it significantly easier to implement this behavior in a compiled model. Similar behavior has been explored in [25], which focused on gating rather than a change to the attention mechanism.

## 3.3 Graph Programs

Graph attention networks naturally resemble algorithmic structure, especially for highly parallel graph algorithms such as Bellman-Ford and Breadth-First-Search (BFS). Importantly, this means that for many algorithms in CLRS, there is an intended ground-truth mechanism, especially the ones we have chosen for our analysis. Our neural compilation method introduces on a domain-specific programming language for specifying programs, which we call graph programs. A graph program consists of multiple components: a variable encoding in the hidden states of the model, an update function for the hidden state, an initialization function, and encoders/decoders. Appendix C contains visualizations of compiled parameters for minimal models.

**Variable Encoding**  Variable encoding structures the hidden vectors, $h$ at each node in terms of named variables. For example, in the Bellman-Ford algorithm, a minimal program needs to track four variables: `visited`, a binary flag indicating if a node has been reached, `distance`, the cumulative distance to reach a node, `id`, the node id, and $\pi$, the predecessor in the shortest path. Note that these are also the variables captured in CLRS traces. They are represented in a vector:

$$h = [\,\texttt{dist visited}\ \pi\ \texttt{id}\,] = [d\,v\,\pi\,x] \tag{15}$$

Then, computing an algorithm is a matter of updating these variables at each timestep according to an update rule. For example, for Bellman-Ford, the update rule for node $i$ with neighbors $j$ is:

$$v = \max(v_i, v_j) \tag{16}$$

$$d = \min(d_j + \mathcal{E}_{ij}) \tag{17}$$

$$\pi = \arg\max_d(x_j) \tag{18}$$

**Update Function**  GATv2 relies on using the attention mechanism to perform computation, in particular using softmax to select among incoming information. We will explain the update function in terms of the graph program for Bellman-Ford (Listing 1). BFS is similar. Note that this is demonstrated for a minimal model with hidden size 4, but our actual model uses the default hidden size of 128. First, the graph attention network must make candidate values $\nu$, which is done by $\underset{\text{val}}{W}$. For example, a compiled $\underset{\text{val}}{W}$ is a sparse matrix that propagates distance (line 9), the visited state (line 10), and permutes an incoming node id $x$ into a potential predecessor variable, $\pi$ (line 11).

$$C_{0:} = [d_0 \quad v_0 \quad 0 \quad x][d_h \quad v_h \quad \pi_h \quad x] = [X_{0:}|H_{0:}] \tag{19}$$

$$\underset{\text{value}}{W} = \begin{bmatrix} 0 & 0 & 0 & 0 & 0 & 0 & 0 & 0 \\ 0 & 0 & 0 & 1 & 0 & 0 & 0 & 0 \\ 0 & 1 & 0 & 0 & 0 & 1 & 0 & 0 \\ 0 & 0 & 0 & 0 & 1 & 0 & 0 & 0 \end{bmatrix} \qquad \underset{\text{value}}{W} C_{0:} = \nu_{0:} = \begin{bmatrix} x \\ \pi_h \\ v_h \\ d_h \end{bmatrix} = \begin{bmatrix} 0 \\ x \\ v_0 + v_h \\ d_h \end{bmatrix} \tag{20}$$

The goal is for softmax to select from these values for the next hidden state. Attention weights are calculated from $\zeta$, which in turn is created by $\underset{\text{in}}{W}$, $\underset{\text{out}}{W}$, and $\underset{\text{edge}}{W}$. Essentially, $\underset{\text{in}}{W}$ propagates the cumulative distance from incoming nodes using a large negative value $-c$, but also masks non-visited nodes by using a large positive value $k$. Then, $\underset{\text{edge}}{W}$ uses large negative values for edge distances. This results in the softmax function receiving values that select for the incoming neighbor with the smallest cumulative distance as a distribution, e.g. $\alpha = [0.01, 0.01, 0.97, 0.01]$ (line 16).

$$\underset{\text{in}}{W} = \begin{bmatrix} 0 & k & 0 & 0 & -c & k & 0 & 0 \\ 0 & k & 0 & 0 & -c & k & 0 & 0 \\ 0 & k & 0 & 0 & -c & k & 0 & 0 \\ 0 & k & 0 & 0 & -c & k & 0 & 0 \end{bmatrix} \qquad \underset{\text{edge}}{W} = \begin{bmatrix} 0 & 0 & 0 & -c \\ 0 & 0 & -c & 0 \\ 0 & -c & 0 & 0 \\ -c & 0 & 0 & 0 \end{bmatrix} \tag{21}$$

The updated hidden state is a weighted combination of candidate values created with $\alpha$. Finally, $\underset{\text{skip}}{W}$ maintains the node's id (line 12), and $\underset{\text{info}}{W}$ adds the edge distance to the cumulative distance (line 9):

$$\underset{\text{skip}}{W} = \begin{bmatrix} 0 & 0 & 0 & 1 & 0 & 0 & 0 & 0 \\ 0 & 0 & 0 & 0 & 0 & 0 & 0 & 0 \\ 0 & 0 & 0 & 0 & 0 & 0 & 0 & 0 \\ 0 & 0 & 0 & 0 & 0 & 0 & 0 & 0 \end{bmatrix} \qquad \underset{\text{info}}{W} = \begin{bmatrix} 0 & 0 & 0 & 0 \\ 0 & 0 & 0 & 0 \\ 0 & 0 & 0 & 0 \\ 1 & 0 & 0 & 0 \end{bmatrix} \tag{22}$$

When present, the pre-attention bias is an identity matrix multiplied by a large positive constant, $k$, indicating that node values should remain the same by default. The attention head itself is a vector of ones, since the important computations have already been done by $\underset{\text{in}}{W}$, $\underset{\text{out}}{W}$, and $\underset{\text{edge}}{W}$. $\underset{\text{meta}}{W}$ is not used.

$$\mathcal{B} = k * I \qquad \omega = \mathbb{1} \qquad \underset{\text{meta}}{W} = \mathbb{0} \tag{23}$$

Appendix C contains visualizations of these parameters. Beyond those presented here, it is also necessary to create encoder/decoder parameters. These have a similar structure to $\underset{\text{val}}{W}$, in that they are often sparse selection matrices or identity matrices (e.g. since $H$ is already a trace).

```
bellman_ford = GraphProgram(
    hidden = HiddenState(
        visit: Component[Bool, 1],
        dist:  Component[Float, 1],
        pi:    Component[Float, 1],
        idx:   Component[Float, 1]
    ),
    update = UpdateFunction( # Function of self, other, init, edge
        dist  = self.dist + edge.dist
        visit = other.visit | self.visit | init.start
        pi    = other.idx
        idx   = self.idx
    ),
    select  = SelectionFunction(
        type = minimum
        expr = other.dist + edge
    )
    mask    = other.visit
    default = self.idx
)
```

Listing 1: Graph Program for Bellman-Ford

# 4 EXPERIMENTS

## 4.1 DEFINING MECHANISTIC FAITHFULNESS

To quantify mechanistic gaps, we compare to a ground-truth reference of the algorithm's behavior. Since there are many correct weight settings that can implement correct behavior, we propose instead comparing behavior within the attention mechanism, which captures abstract learned behavior. In the GATv2 architecture, the attention mechanism specifies how information should transfer between nodes. This tightly constrains expected attention mechanism behavior: exploring frontier nodes for BFS, selecting between minimum incoming paths for Bellman-Ford, swapping items in bubble sort, and so on must all use attention carefully. Furthermore, the correct attention patterns can be created by a variety of internal weight settings and hidden-state structures, which is how it captures abstract behavior. Even though comparing in attention space eliminates a lot of issues with comparing in weight space, we also compare across 128 random initializations. This allows us to diagnose mechanistic failures at a systematic level, and eliminate the choice of ground-truth as a factor.

Furthermore, we validate attention-based mechanistic faithfulness by measuring trace prediction accuracy, a built-in capability of the CLRS benchmark. We call this external faithfulness, because if the learned algorithm is correct, it should correctly predict the trace regardless of the internal details of how it is implemented. These definitions result in two quantitative faithfulness measures:

**Internal Faithfulness** considers the timeseries of attention states, $\alpha$, and compares learned mechanisms $\hat{\alpha}$ to a ground truth $\alpha^\star$, using an L1 norm that sums across the time and two node axes.

$$\phi_{\text{internal}} = \frac{|\hat{\alpha} - \alpha^\star|}{t * n * n} \tag{24}$$

**External Faithfulness** measures average accuracy over a timeseries of predicted traces. Traces contain different types of predictions, $y$: numerical (e.g. cumulative distance), binary predictions (e.g. if a node has been reached), and class predictions (e.g. a parent node id). These are evaluated within a margin, ($\epsilon = \{0.5, 0.1, 1e{-}6\}$, respectively) to convert them to binary matching scores, and then the timeseries of matches is averaged. $\mathbb{1}$ represents the indicator function, and $t$ the length of the trace.

$$\phi_{\text{external}} = \frac{\sum \mathbb{1}(\hat{y} - y^\star < \epsilon)}{t} \tag{25}$$

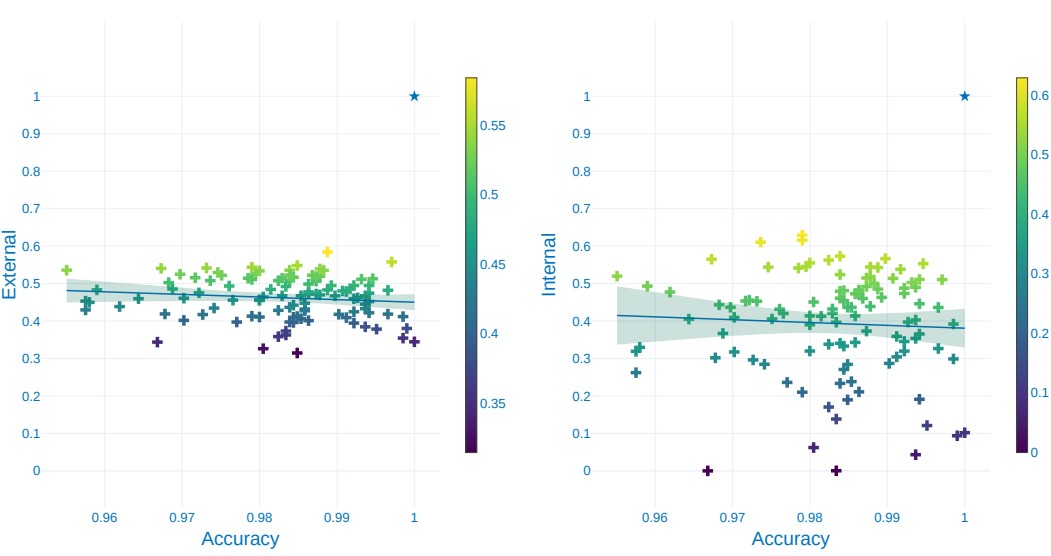

Figure 1: External and Internal Faithfulness of BFS

## 4.2 FAITHFULNESS

We measure both internal and external faithfulness, and find that there is no significant correlation between faithfulness and accuracy (Figures 1 and 10, 95% confidence interval, Table 1).

| Measure | Pearson $r$ | $p$-value | Spearman $\rho$ | $p$-value |
|---|---|---|---|---|
| External | -0.124 | 0.203 | -0.130 | 0.178 |
| Internal | -0.055 | 0.569 | -0.018 | 0.856 |

Table 1: Correlation Coefficients between effectiveness and faithfulness measures for learned BFS

### 4.3 INTERNAL FAITHFULNESS

Figure 2 shows internal faithfulness (Equation 24) over time for BFS. Figure 3 visualizes the closest matching attention trace from the sampled initializations. The attention mechanism is slightly closer at the beginning of computation (in this case, the first two steps), but deviates after this. However, the observed mechanistic gaps, visualized in Figures 2 and 3, are quite large, far beyond the amount that would be explained by factors like tie-breaking or attention sharpness. In combination with the number of initializations tested, this indicates a systematic failure to learn faithful internal mechanisms, even when predictive accuracy is nearly perfect.

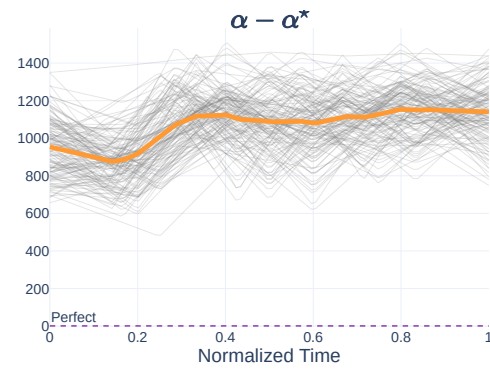

Figure 2: BFS Attn. Timeseries

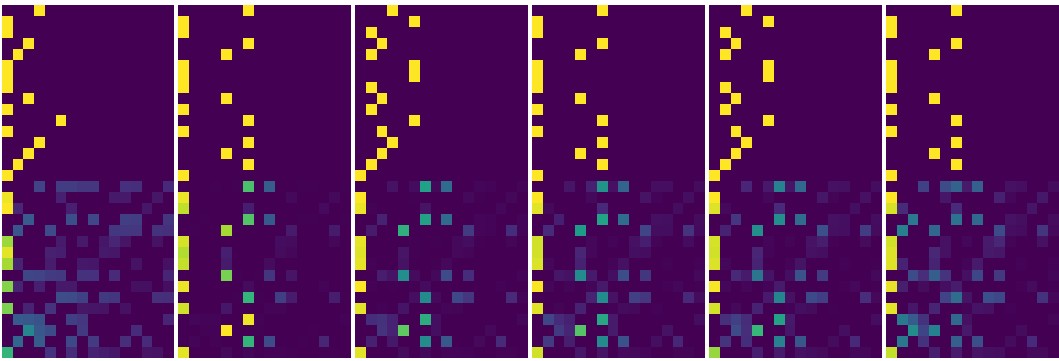

Figure 3: BFS Attention Mechanism Comparison (Best Match)

### 4.4 EXTERNAL FAITHFULNESS

We plot trace predictions on a uniform timescale, showing how they are only partially consistent and degrade over time (Figure 4). Notably, this behavior occurs even on the training and validation sets. Inconsistent trace predictions validate our findings around internal faithfulness.

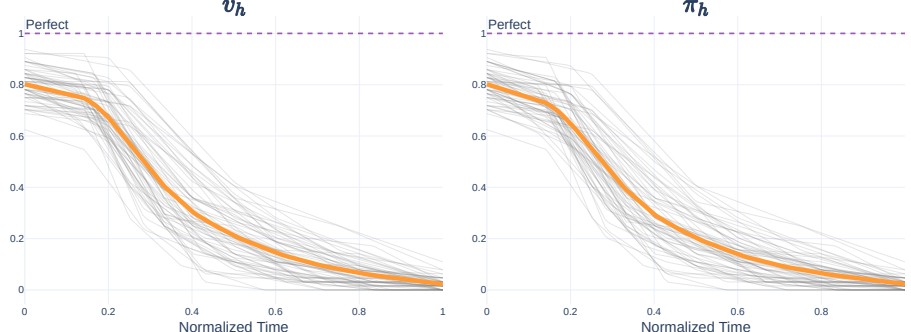

Figure 4: Learned BFS Trace Predictions Over Time

## 4.5 Validation of Faithfulness Metrics

To ensure our analysis of internal faithfulness is valid, we want to ensure that comparisons to a particular compiled solution are not arbitrary. First of all, attention captures abstract learned algorithmic behavior, and already constraints the set of correct behaviors significantly, since different hidden state structures or exact parameter settings can still produce the same attention patterns. In GATv2, the only way for information to flow between nodes is via the attention mechanism, and under default settings, there is only a single attention head. This implies that for algorithms to be learned faithfully in GATv2, they must use the attention mechanism in the intended way.

One of the main ways an equally correct algorithm could differ is in tie-breaking, but the CLRS benchmark specifies an arbitrary tie-breaking preference in terms of node position [1]. Another consideration is attention sharpness [48, 49], e.g. where the softmax ranking is correct, but some probability is assigned to incorrect locations. However, a fully correct learned algorithm will have a sharp distribution, and solutions that are nearly-correct but not sharp will not be penalized significantly by Equation 24. There is also the possibility of behavior that occurs outside of the attention mechanism, but we consider these to be mechanistic failures, since in general, it is not possible to implement a correct non-trivial algorithm in GATv2 without using the attention mechanism.

**Inter-Solution Comparison** Beyond these considerations, we utilize a large number of 128 random initializations, finding that none of them exhibit behavior similar to the compiled reference, indicating that our results are not influenced by the choice of ground-truth mechanism. We also compare attention patterns within the set of learned solutions across different initializations. Figure 5 shows how each of the learned solutions to BFS compares to each other, and visualizes clusters as a dendrogram under ward linkage, demonstrating the diversity of learned solutions, with 18 consistent clusters around suboptimal solutions. On average, inter-solution differences are 0.009, but the closest distance to the compiled solution is 0.37 in terms of Equation 24.

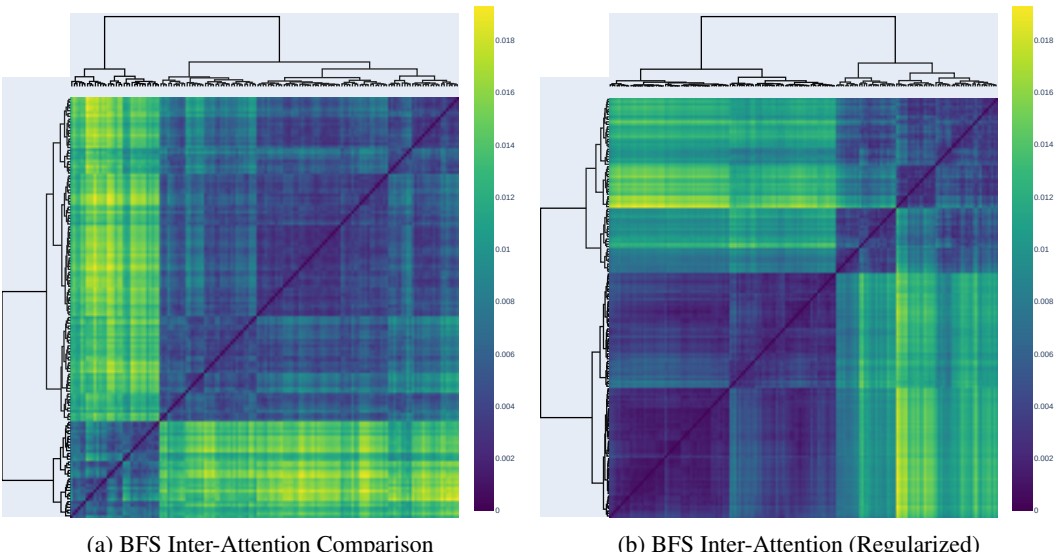

(a) BFS Inter-Attention Comparison         (b) BFS Inter-Attention (Regularized)

Figure 5: Inter-Attention Comparison Between Learned Solutions for BFS

## 4.6 Potential Causes of Mechanistic Gaps and Their Implications for NAR

Mechanistic gaps imply the potential for improving NAR models. Specifically, our analysis clarifies that even if predictive accuracy is high, it is possible for internal mechanisms of a model to be underconverged or underutilized. We believe that underconvergence on traces is the main cause of our observations, since the learned models (under default settings) have not converged to accurately predict traces even in the training set. This indicates that the default training settings do not cause hidden states to be properly learned. One potential solution to this is to adopt more structured curriculum learning, e.g. where predicting traces is prioritized before predicting the answer is.

Within GATv2 specifically, we also outlined specific architectural changes that can prohibit learning proper mechanisms, e.g. the way edge information is utilized. Previous work has also explored this [25], but we also confirm that our modifications partially alleviate this gap (Appendix D.4).

Table 2: Ablation: Bellman-Ford Edge Information (Mean $\pm$ Stddev (Max))

| Experiment | Performance |
| --- | --- |
| Default (No Edge Info) | $86.59\% \pm 5.97\%(92.24\%)$ |
| Modified (Edge Info) | $90.67\% \pm 1.40\%(92.72\%)$ |

Other explanations for mechanistic failures include the scalar bottleneck hypothesis [43, 44], lottery ticket hypothesis [50], and algorithmic phase space hypothesis [10]. Neural networks do not naturally learn sparse interpretable algorithms that match expected mechanisms. Instead, it's highly likely that they learn multiple partial solutions in parallel and combine them [39, 51, 41]. Under these hypotheses, then learning mechanistically faithful algorithms requires much more sophisticated training procedures.

## 5  CONCLUSION

In this paper, we propose measures of mechanistic faithfulness, with the aim of building neural algorithmic reasoning systems that produce more general and robust solutions. Specifically, we introduce a neural compilation method for compiling algorithms into graph attention networks, and then use the intermediate attention states of the compiled model as a reference for ideal behavior. In doing so, we establish mechanistic gaps, even for BFS, which GATv2 is algorithmically aligned to.

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

# A  Training Details

For training, we use the unaltered CLRS dataset and default hyperparameter settings (which have been well-established by previous literature). For optimization, we use the humble adam optimizer [52]. We use the hyperparameters reported in Table 8. For additional experiments, we use the following settings, derived from the defaults on the right:

Table 3: Settings for Trace Ablation

| hint_mode | none |
|---|---|

Table 4: Settings for Minimal Experiments

| hidden_size | 4 |
|---|---|

Table 5: Settings for Regularization Experiments

| regularization | True |
|---|---|
| regularization_weight | {1.0000e−3, 1.0000e−4} |

Table 6: Settings for Grokking Experiment

| train_steps | 50000 |
|---|---|
| learning_rate | 5.0000e−5 |

Table 7: Settings for Architecture Ablations

| train_lengths | 16 |
|---|---|
| simplify_decoders | True |
| use_edge_info | {True,False} |
| use_pre_att_bias | {True,False} |
| length_generalize | False |

| algorithms | bellman_ford |
|---|---|
| train_lengths | 4, 7, 11, 13, 16 |
| random_pos | True |
| enforce_permutations | True |
| enforce_pred_as_input | True |
| batch_size | 32 |
| train_steps | 10000 |
| eval_every | 50 |
| test_every | 500 |
| hidden_size | 128 |
| nb_heads | 1 |
| nb_msg_passing_steps | 1 |
| learning_rate | 1.0000e−4 |
| grad_clip_max_norm | 1.0000 |
| dropout_prob | 0.0000 |
| hint_teacher_forcing | 0.0000 |
| hint_mode | encoded_decoded |
| hint_repred_mode | soft |
| use_ln | True |
| use_lstm | False |
| encoder_init | xavier_on_scalars |
| processor_type | gatv2 |
| freeze_processor | False |
| simplify_decoders | False |
| use_edge_info | False |
| use_pre_att_bias | False |
| length_generalize | True |
| regularization | False |
| regularization_weight | 1.0000e−4 |
| git hash | 445caf85 |

Table 8: Settings for Trained Bellman-Ford

# B  Extended Methods

## B.1  Encoders and Decoders

Beyond the parameters and equations presented above, a graph attention network has additional layers for encoding and decoding. Often, they are simply linear layers that produce vector representations of input data or traces. Effectively, the input vector $v_l$ is a function of multiple encoders, e.g. for raw inputs $\hat{v}$ (representing different graph features or input traces), the encoded input is:

$$v_l = \underset{\text{enc}}{W_{lk}}\, \hat{v}_k \tag{26}$$

Furthermore, a graph attention network may have multiple outputs, for instance different trace predictions for various algorithms. Each of these has a separate problem-specific decoder. In more complex cases, answers are decoded using multiple layers, involving the edge features $\mathcal{E}$:

$$p_1 = W_1\, h \quad p_2 = W_2\, h \quad p_e = W_e\, \mathcal{E} \tag{27}$$
$$p_m = \max(p_1, p_2 + p_e) \tag{28}$$
$$y = W_3 p_m \tag{29}$$

We note this level of detail because it is critical for understanding the behavior of the learned models: A surprising amount of computation is happening in the decoding layers. Also, compiling algorithms into graph attention networks is not only a matter of setting the weights of the main graph attention parameters, but also the parameters of the encoders and decoders.

## C  GRAPH PROGRAMS

A graph program consists of two components: a variable encoding in the hidden states of the model, and a compiled update function that updates the hidden state. Since hidden states begin uninitialized, the update function is also responsible for setting them in the initial timestep. The core of the update function relies on using the attention mechanism to perform computation. Fundamentally, this is a matter of using the GNN's aggregation function, in this case softmax. Specifically, the inputs to softmax allow computing a max or min, or masking based on boolean states.

Both Bellman-Ford and BFS use softmax to compute a minimum, but Bellman-Ford does so over cumulative distance, while BFS does so over node id order. In both algorithms, the visitation status of each node is used to mask attention coefficients, defaulting to self-selection.

```
bfs = GraphProgram(
    hidden = HiddenState(
        s:    Component[Bool, 1],
        pi:   Component[Float, 1],
        idx:  Component[Float, 1]
    ),
    update = UpdateFunction( # Function of self, other, init, edge
        visit = other.visit | self.visit | init.start
        pi    = other.idx
        idx   = self.idx
    ),
    select  = SelectionFunction(
        type = minimum
        expr = other.idx
    )
    mask    = other.visit
    default = self.idx
)
```

Listing 2: Graph Program for BFS

## C.1 COMPILED BELLMAN-FORD

For example, in Bellman-Ford, the attention mechanism selects edges based on cumulative distance. In Figure 6, $W_{\text{edge}}$ contains large negative values on the diagonal, which forces attention to select strongly based on edge distance. However, because node-expansions are only valid along the frontier, large positive values in $W_{\text{in}}$ and $W_{\text{out}}$ control the attention mechanism to default to retaining hidden states when nodes aren't valid for expansion (using $\mathcal{B}$, labelled $W_{\text{pre\_attn\_bias}}$). Similarly, the negative values in $W_{\text{in}}$ add cumulative distance for the attention mechanism. Weight settings in $W_{\text{skip}}$ and $W_{\text{value}}$ create and maintain structured hidden vectors. Specifically, the hidden vector representation is:

$$h = [\,\text{dist visited } \pi \text{ id}\,] \qquad (30)$$

In this case, the first component of $h$ contains cumulative distance (maintained also by $W_{\text{edge 2}}$). The second component of $h$ indicates if a node has been reached, the third component corresponds to the predecessor node in the path, and the fourth component of $h$ encodes the node's id.

Finally, the attention head $W_{\text{a\_0}}$ simply accumulates attention values using a vector of all ones. Note that these parameters are for the *minimal* version of Bellman-Ford, using a tiny 500-parameter network with a size 4 hidden state. We have generalized this to larger networks, e.g. the size 128

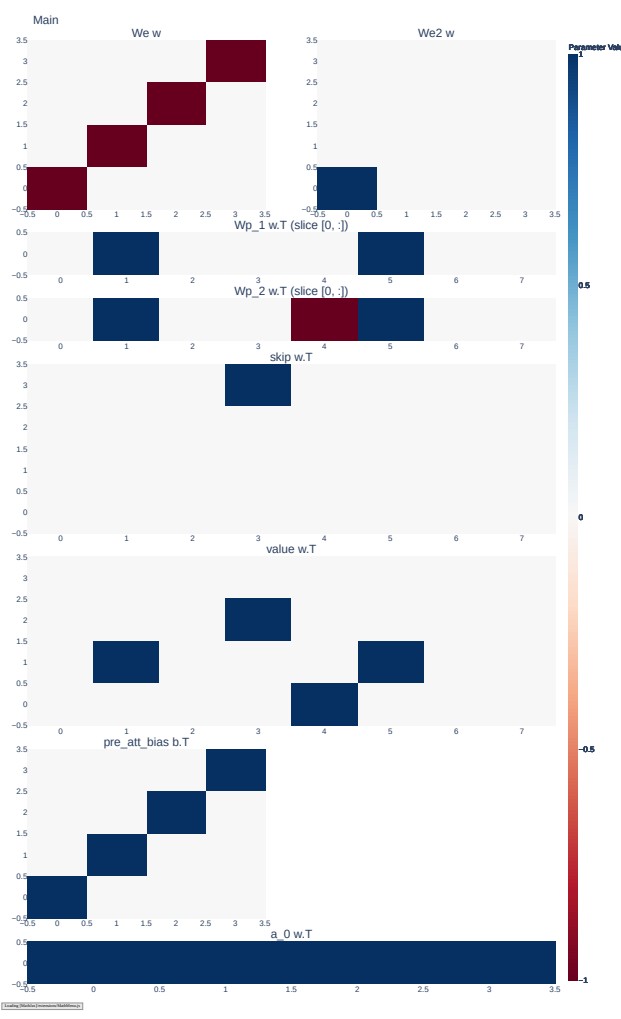

Figure 6: Main Parameters for Bellman-Ford

hidden state model that matches the dimensions of GNNs trained in the CLRS benchmark, which has about 5e6 parameters. This is a matter of extending the patterns shown in Figure 6.

These parameter values are the *output* of a compiled graph program. Since Bellman-Ford was the first algorithm we compiled, before we developed the graph program language, the values were set by hand. However, each parameter value corresponds to a part of a graph program. The first part of the graph program establishes Equation 30, setting these based on inputs. Then, the graph program update function describes state-maintenance and the attention update, which compiles into $W_{\text{edge}}$ $W_{\text{in}}$ $W_{\text{out}}$ $W_{\text{pre\_attn\_bias}}$ $W_{\text{in}}$ $W_{\text{skip}}$ $W_{\text{value}}$ $W_{\text{edge 2}}$ and $W_{\text{a\_0}}$.

To fully implement Bellman-Ford, it is also necessary to modify the parameters of encoders and decoders, with relevant parameter settings shown in Figure 7. For encoders, like $W_{\text{enc\_s}}$, they are sparse vectors that place relevant information (in this case, which node is the starting location) Since node ids are stored as linear positional encodings, they must be decoded into one-hot classifications, which is the role of $\pi_{\text{dec}}$. These simply use the equation:

$$y_{\text{pred}} = \texttt{softmax}(c \cdot \max(p - v, v - p)) \qquad (31)$$

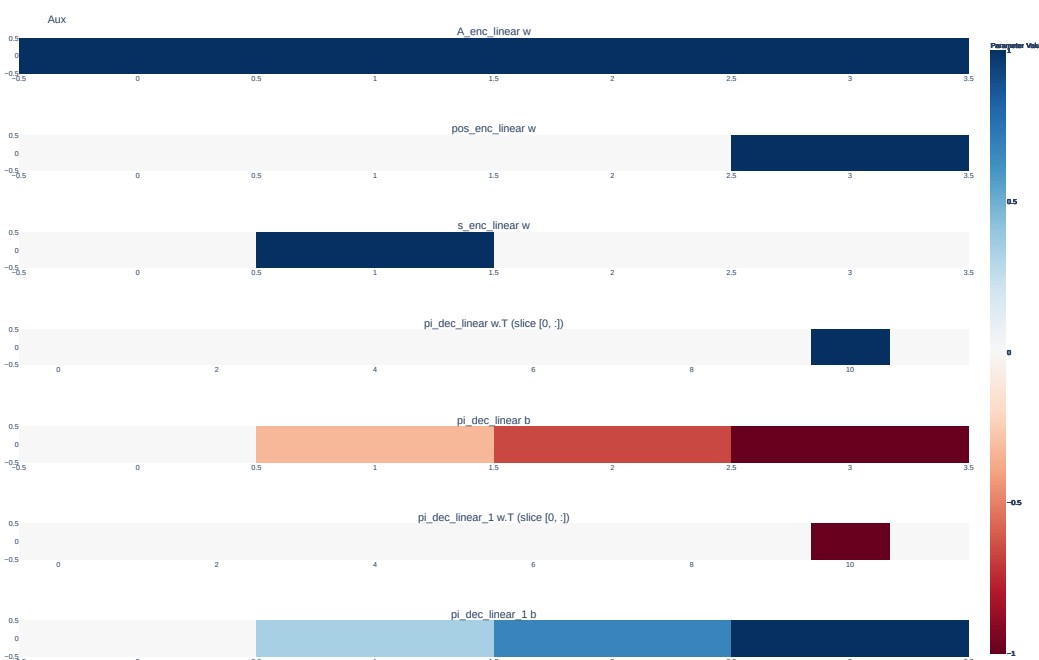

Figure 7: Auxilliary Parameters for Bellman-Ford

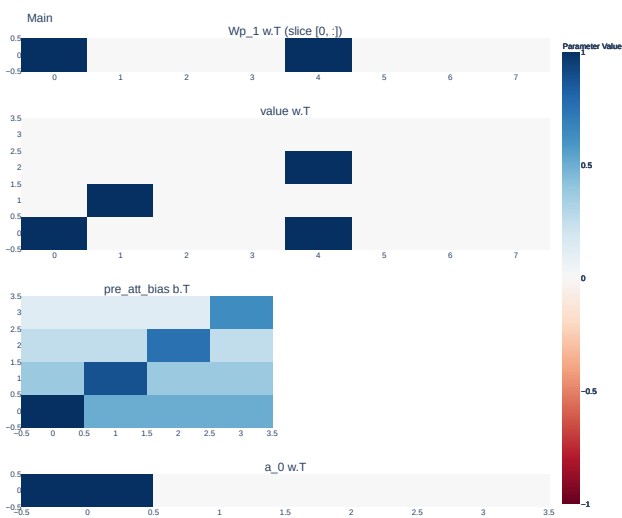

Where $v$ is the positional encoding, $p$ is a vector of all positional encodings, and $c$ is a large negative constant, e.g. $-1e3$. For instance if $v = [0.25]$, $p = [0.0\ 0.25\ 0.5\ 0.75]$, then $y = [0\ 1\ 0\ 0]$. Using positional encodings throughout the model prevents the need for having unwieldy one-hot encodings as a core part of the architecture, reducing the overall parameter count and improving numerical stability. However, it also introduces a scalar bottleneck, since the individual components of $h$ each contain critical information.

Figure 8: Main Parameters for BFS

## C.2   COMPILED BFS

Compiling BFS is largely similar to compiling Bellman-Ford, with the only notable difference being that cumulative distances are never tracked, and the pre-attention bias $\mathcal{B}$ plays two roles: First, it biases towards self-selection, e.g. when a node is not being expanded, its state remains the same. Second, it biases towards expanding nodes with lower ids, for instance if a is adjacent to both b and c, then the edge a-b is added, but a-c is not. Otherwise, the main parameters and encoder parameters are largely identical to those for Bellman-Ford.

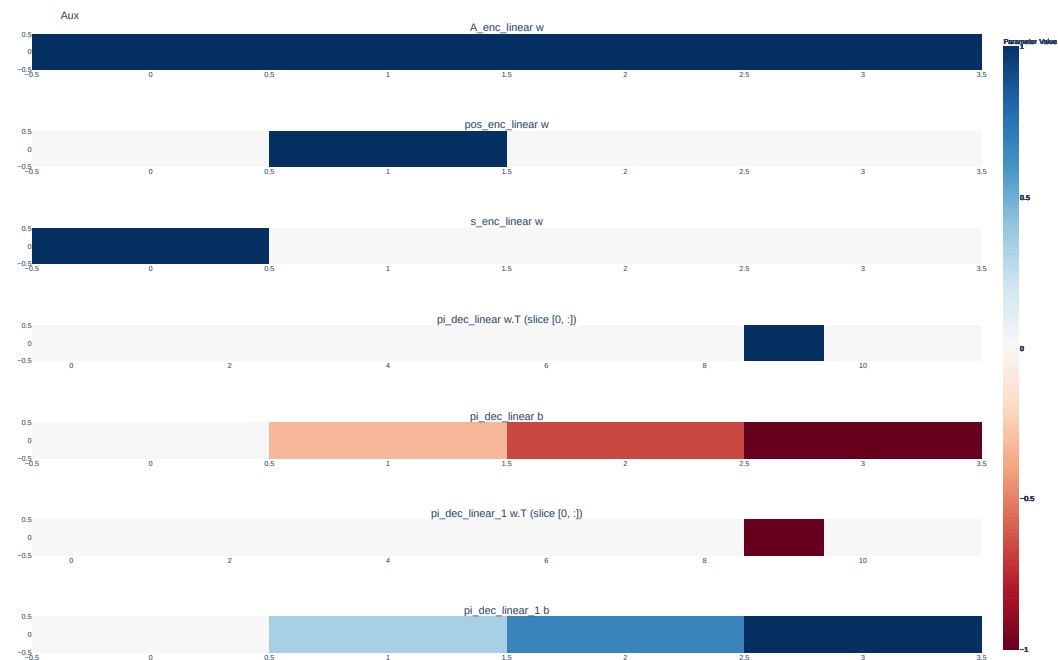

Figure 9: Auxilliary Parameters for BFS

Table 9: Regularization, Grokking, and Minimal experiments

| Algorithm | Regularization | Extended Training | Minimal |
|---|---|---|---|
| BFS | $97.76\% \pm 1.05\%$ | $97.55\% \pm 1.52\%$ | $81.60\% \pm 11.32\%$ |
| Bellman-Ford | $87.35\% \pm 1.68\%$ | - | $87.35\% \pm 1.68\%$ |

## D   EXTENDED RESULTS

### D.1   GROKKING, REGULARIZATION, AND MINIMAL MODELS

### D.2   BELLMAN-FORD EXTERNAL AND INTERNAL FAITHFULNESS

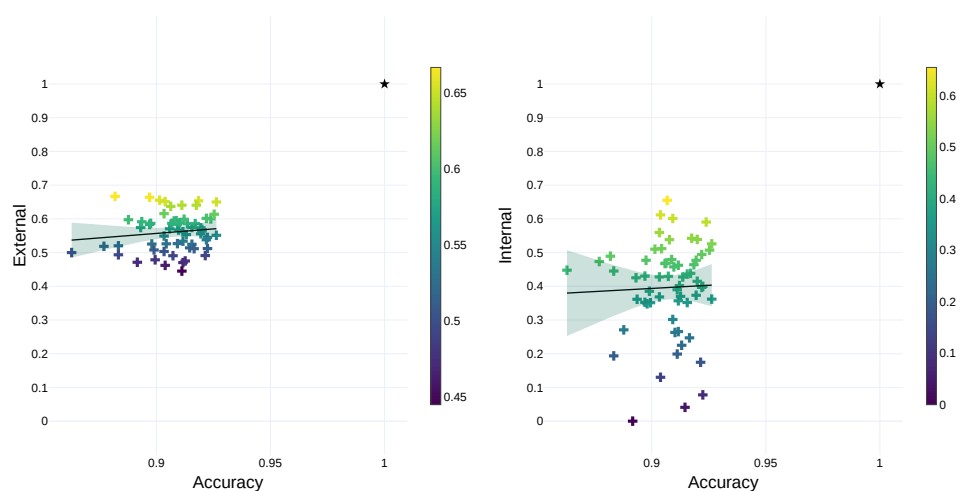

Figure 10: External and Internal Faithfulness of Bellman-Ford

## D.3 BASELINES

We replicate baseline results with a sampling budget of 128 initializations. This provides a variety of solutions to compare against, ensuring that initializations do not confound our analysis [50]. We use CLRS benchmark default hyperparameters. See our dedicated Appendix A for full settings.

Table 10: Replicated GATv2 Baseline CLRS Results (Mean $\pm$ Stddev (Max))

| BFS | DFS | Bellman-Ford |
|---|---|---|
| $98.30\% \pm 0.97\% (100.00\%)$ | $12.74\% \pm 3.44\% (18.21\%)$ | $90.63\% \pm 1.27\% (92.77\%)$ |

## D.4 ARCHITECTURE ABLATIONS

In Section 3.2, we introduce two modifications to the graph attention network architecture, namely introducing edge information (specifically for Bellman-Ford), and introducing a pre-attention bias matrix (for both Bellman-Ford and BFS). Of these two changes, the introduction of edge information is potentially more interesting, as it reveals a potential architecture-level reasoning that the learned version of Bellman-Ford may not be faithful. However, the change is not strictly necessary to be able to compile Bellman-Ford, but it certainly makes compiling the algorithm significantly easier, and closer to the intended faithful behavior. Adding the pre-attention bias is also not strictly necessary, but makes it more natural to control each algorithm's default behavior.

**Edge Information**   We hypothesize that the learned version of Bellman-Ford may be struggling partially because it cannot track cumulative path distances in a faithful way. If this were the case, then we would expect the unmodified architecture to perform worse than the modified one, assuming that learning is capable of exploiting this architecture change in the way that we expect. However, it may be the case the without the architecture change, the model is able to track cumulative edge distances by leaking information through the attention mechanism, or by delaying cumulative path length calculation to the decoding step.

Table 11: Ablation: Bellman-Ford Edge Information (Mean $\pm$ Stddev (Max))

| Experiment | Performance |
|---|---|
| Default (No Edge Info) | $86.59\% \pm 5.97\% (92.24\%)$ |
| Modified (Edge Info) | $90.67\% \pm 1.40\% (92.72\%)$ |

In Table 11, we find that, while maximum performance is unaffected, the learning algorithm is more commonly able to find high-quality solutions, bringing up the average performance, and reducing the standard deviation between solutions.

**Pre-Attention Bias**   Unlike introducing edge information, adding a pre-attention bias is less necessary for the model to learn correct behavior. However, within the learned parameters, each bias matrix can only the pre-attention values, $\zeta$ on either the row or column axis, but cannot bias unaligned components, such as having an identity matrix as a bias (which is needed for compiled BFS). A major downside of introducing a pre-attention bias is that its size is tied to problem size, preventing length-generalization, which outweighs the benefits of introducing it.

Table 12: Ablation: BFS Pre-Attention Bias (Mean $\pm$ Stddev (Max))

| Experiment | Performance |
|---|---|
| Default (Without Bias) | $99.92\% \pm 0.28\% (100.00\%)$ |
| Modified (With Bias) | $99.72\% \pm 0.95\% (100.00\%)$ |

Since the baseline performance of BFS is so high, Table 12 does not show significant differences, possibly because the results are within distribution (tested on length 16). Next, we try introducing both

modifications to a length-limited version of Bellman-Ford. However, the lack of length generalization makes the results difficult to interpret, but at the very least the model is still as-capable as the unmodified version within distribution.

Table 13: Ablation: Bellman-Ford Both (Mean $\pm$ Stddev (Max))

| Experiment | Performance |
| --- | --- |
| Default (Neither) | $97.31\% \pm 0.92\%(98.93\%)$ |
| Modified (Both) | $97.81\% \pm 0.88\%(99.41\%)$ |

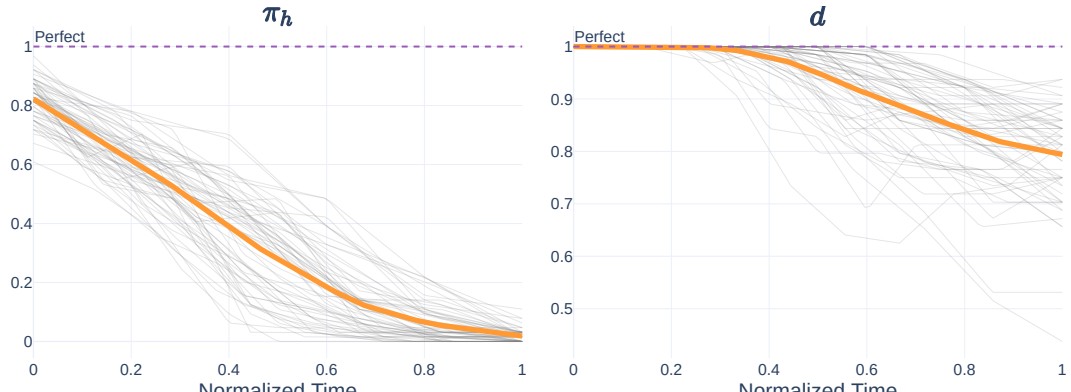

Figure 11: Learned Bellman-Ford Trace Predictions Over Time

## D.5  ADDITIONAL RESULTS ON TRACES

Trace faithfulness also affects BFS, which, even though it is highly effective, quickly deviates in predicting traces (Figure 11). This behavior is curious, as BFS is high-performing, so conceivably it has learned to track whether each node has been reached. It's possible the issue is less about internal representation, and more about the ability to decode internal representations back into trace predictions.

## D.6  TRAINING WITHOUT TRACES

While it may seem that intermediate traces are critical in learning algorithms faithfully, there are many cases where they are not necessary or even hurt performance [1, 53].

Table 14: Training Without Traces (Mean ± Stddev (Max))

| Experiment | Performance |
|------------|-------------|
| DFS | $16.49\% \pm 2.45\% (20.61\%)$ |
| BFS | $98.74\% \pm 0.98\% (100.00\%)$ |
| BF | $90.14\% \pm 1.15\% (91.80\%)$ |

## D.7  MINIMAL EXPERIMENTS

Our neural compilation results establish that a 500-parameter GAT can express BFS or Bellman-Ford. While we do not strongly expect gradient descent to find the perfect solutions, we experiment with training minimal models over a large number of random seeds (1024), to see if we draw lucky "lottery tickets" [50]. The results in Table 15 establish that finding high-quality solutions in this regime is possible, but furthermore show that the architecture modifications have a stronger effect on minimal models, which are very constrained by scalar bottlenecks.

Table 15: Minimal Networks (Mean ± Stddev (Max))

| Experiment | Performance |
|------------|-------------|
| Bellman-Ford (Default) | $38.97\% \pm 8.35\% (59.13\%)$ |
| Bellman-Ford (Arch Modify) | $74.38\% \pm 10.29\% (88.77\%)$ |
| BFS (Default) | $81.60\% \pm 11.32\% (99.56\%)$ |
| BFS (Pre-Attention Bias) | $93.32\% \pm 6.89\% (99.32\%)$ |

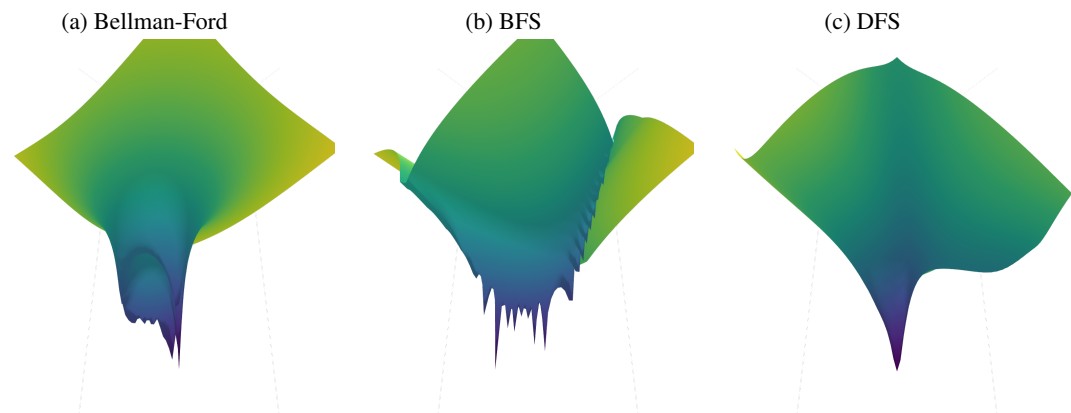

(a) Bellman-Ford        (b) BFS        (c) DFS

Figure 12: Loss Landscapes

## D.8 STABILITY

Beyond comparing learned and compiled solutions, we want to characterize the loss landscapes surrounding compiled minima, and also understand how they are affected by further optimization. We plot this using the technique introduced in [54], which plots gaussian perturbations in terms of two random vectors which have been normalized to be scale invariant.

For example [36] compiled a logic algorithm into the transformer architecture which was both difficult to find and diverged when trained further. We find similar behavior, but it is dependent on random data sampling order, see Appendix D.8. The compilation strategy reported in this paper uses sparse weights, which are affected by the scalar bottleneck and do not resemble learned solutions. Because of the artificial nature of compiled solutions, we expect the minima to be unstable, but hope to use the results of these experiments to inform more sophisticated methods for compiling algorithms into neural networks. We find that compiled solutions, when further trained, can deviate from optimal parameters (Table 16). However, this is highly dependent on data sampling order, resulting in high variance in performance. This indicates that compiled minima are unstable. However, this training is done with mini-batch gradient descent, which is inherently noisy (intentionally). We also attribute these results to the scalar bottleneck hypothesis.

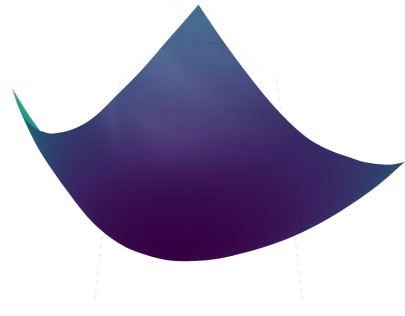

(a) BFS Learned

Table 16: Stability (Mean $\pm$ Stddev (Max))

| Experiment | Performance |
|---|---|
| Compiled $\rightarrow$ Trained Bellman-Ford | $80.77\% \pm 14.83\%(97.66\%)$ |
| Compiled $\rightarrow$ Trained BFS | $82.04\% \pm 15.55\%(100.00\%)$ |

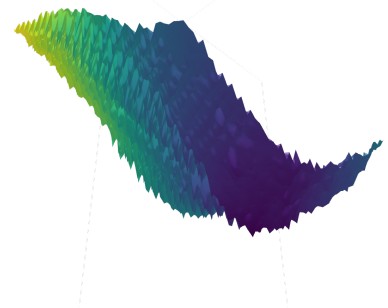

(b) BFS Compiled

Figure 13: Landscape

## D.9 ATTENTION MECHANISM

Figure 14: Bellman-Ford Attention (Full)          Figure 15: BFS Attention (Full)

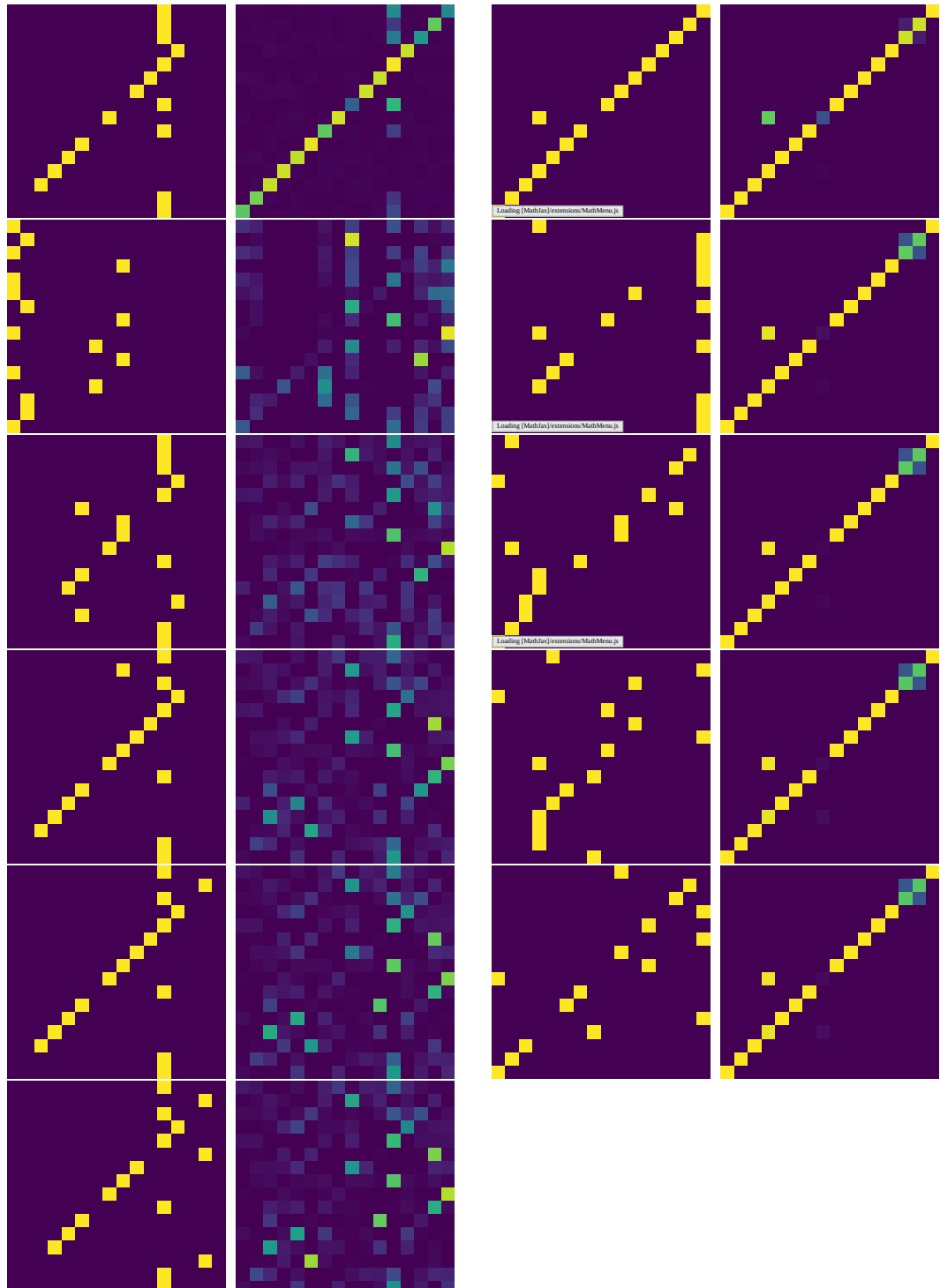

## D.10    DISPARITY BETWEEN BFS AND DFS

To establish that BFS is algorithmically aligned, we explicitly test variants of BFS and DFS so that we can eliminate the confounding variables of trace length and trace complexity.

**Trace Length**    First, because DFS is sequential, the traces used in learning DFS are naturally longer than those for learning BFS. To mitigate this, we create a version of BFS with sequential traces, where rather than expanding all neighbors at once, one neighbor is expanded at a time. The semantics and underlying parallel nature of the algorithm are unchanged, but the traces used for training are artificially made sequential to mimic the long traces used in learning DFS. We find that, even with significantly longer traces, BFS is still significantly more trainable than DFS.

**Trace Complexity**    Second, because DFS requires more sophisticated state tracking, we explicitly test versions of DFS that provide only the most critical information in each trace. By default, DFS traces include predecessor paths, node visitation state, node visitation times, the current node stack, and the current edge being expanded. In the simplified version, we train on only predecessor paths and node visitation state, ignoring times, the node stack, and edge. This more closely resembles the data that BFS is trained on, which also includes only predecessor paths and node visitation state. Later, we experiment with training all algorithms without traces entirely, and also evaluate the effectiveness of learned algorithms at predicting intermediate traces.

Table 17: BFS-DFS Disparity (Mean $\pm$ Stddev (Max))

| Experiment | Performance |
|---|---|
| Sequential BFS | $92.90\% \pm 2.85\%(95.61\%)$ |
| Simplified DFS | $11.66\% \pm 4.16\%(20.75\%)$ |

### D.11   LOSS LANDSCAPES

To better understand the nature of compiled solutions, we plot both the loss landscapes around compiled minima, learned minima, and initialized parameters. We hope to gain insight into the stability of compiled solutions, in particular if they resemble learned ones.

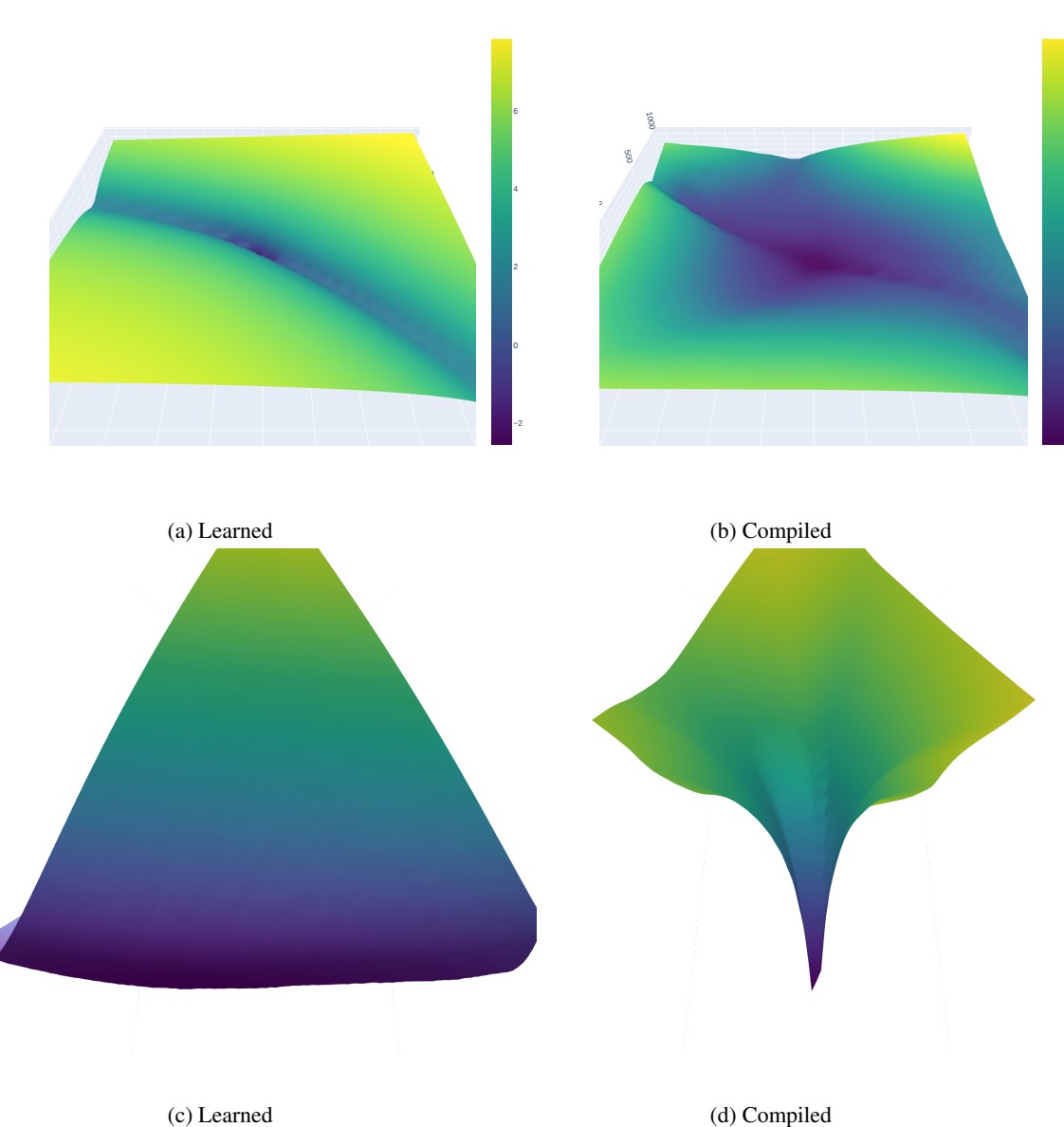

(a) Learned                                 (b) Compiled

(c) Learned                                 (d) Compiled

Figure 16: Bellman-Ford Learned vs Compiled Loss Landscapes (General on Top, Local on Bottom)

**Bellman-Ford Learned vs Compiled Loss Landscapes**   For Bellman-Ford, we find that the loss landscape for the learned solution is flatter and more forgiving than the compiled solution.

**BFS Learned vs Compiled Loss Landscapes**   For BFS specifically, we find that learned solutions have found an extremely flat minima (Figure 17), indicating a high-quality solution (even though it is not faithful). This is not the case for the compiled solution!

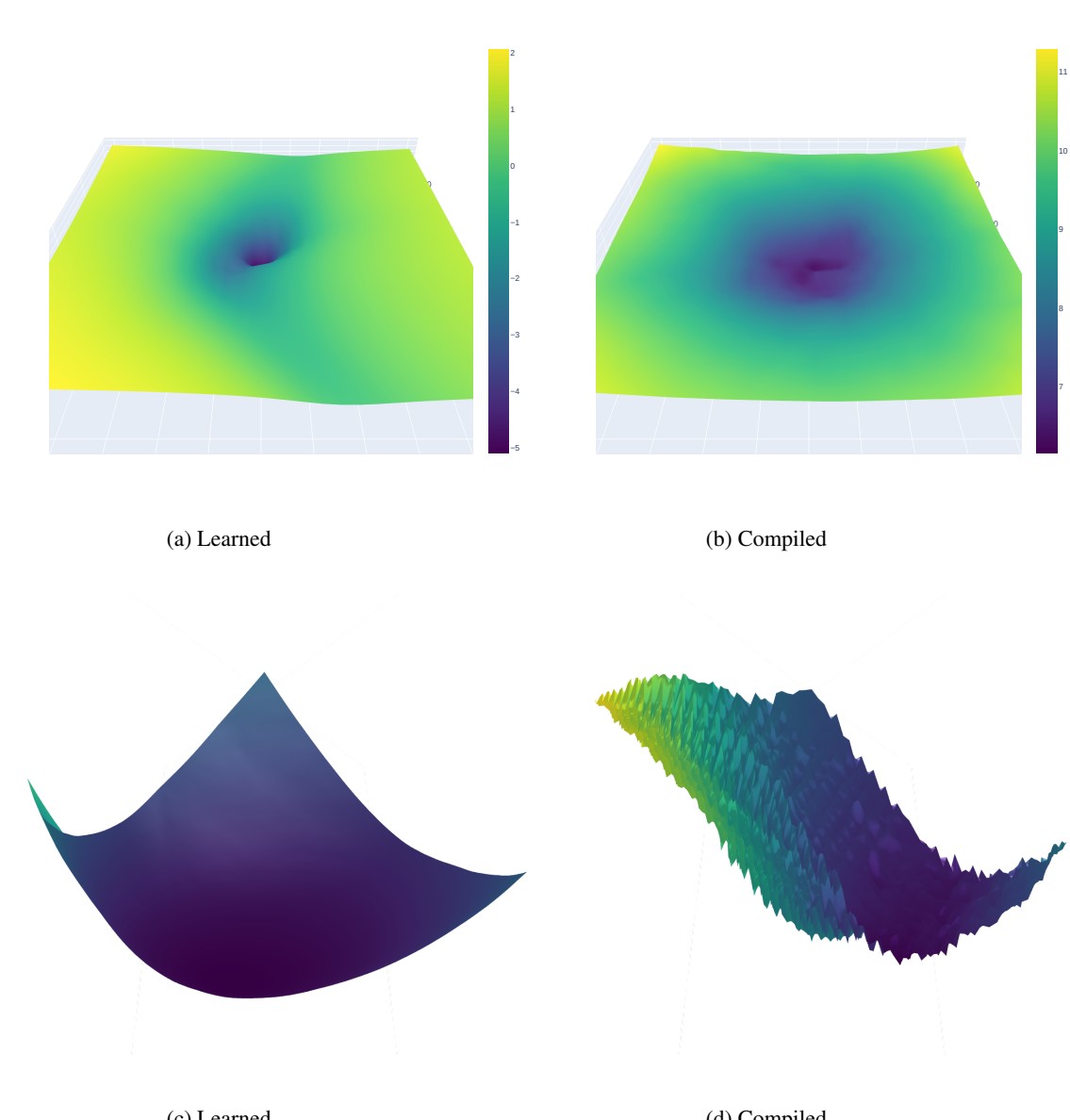

(a) Learned

(b) Compiled

(c) Learned

(d) Compiled

Figure 17: BFS Learned vs Compiled Loss Landscapes (General on Top, Local on Bottom)

**DFS Loss Landscapes**  We cannot draw strong conclusions from the loss landscapes for DFS, but we report them for completeness:

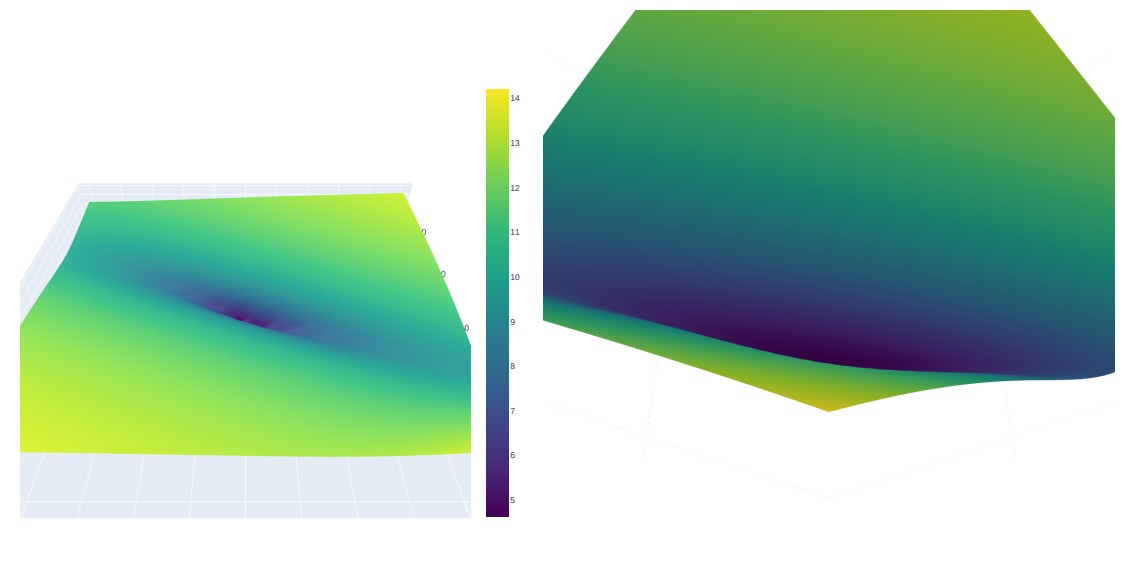

(a) General DFS                                      (b) Local DFS

Figure 18: DFS: Local vs General Landscape

