# OpenReview forum: "Quantifying Mechanistic Gaps in Algorithmic Reasoning via Neural Compilation"
_ICLR.cc/2026/Conference — Submitted to ICLR 2026_

### Official Review · Reviewer_dBJN · 2025-10-20

**Soundness:** 2
**Presentation:** 2
**Contribution:** 2
**Rating:** 2
**Confidence:** 4

**Summary:**

The paper proposes using neural compilation with a graph program to encode source algorithms directly into network parameters as a form of ground-truth, expected behavior. It then compares these compiled attention patterns with those learned by GATv2 when simulating graph algorithms. Two faithfulness measures are introduced: an external one, which evaluates alignment with algorithmic traces, and an internal one, which compares the expected attention maps derived from the graph program with those learned by the model. The paper shows that even for BFS, where GAT achieves near-perfect accuracy, there remains a noticeable gap between the expected and learned attention matrices, suggesting that high task accuracy does not necessarily imply mechanistic faithfulness.

**Strengths:**

- The paper tackles an important and timely topic. While neural algorithmic reasoning has achieved strong performance across a range of algorithms and architectures, few works have examined its interpretability. Understanding this dimension is crucial for developing more robust solutions, and could even inform algorithmic discovery.

- The use of neural compilation as a tool for mechanistic interpretability is interesting, and its application to graph neural networks and algorithmic reasoning tasks is novel.

- The paper is generally well-written, with clear explanations and a thorough review of relevant literature and background material.

**Weaknesses:**

- The paper builds upon the assumption that the attention mechanism defined by the graph program serves as the ground-truth reference for measuring faithfulness. While this provides a basis for comparison, it represents one possible realization of the underlying algorithm rather than a unique solution. This could partly explain why the GNN achieves near-perfect accuracy while exhibiting differences in its learned attention patterns, as it may have discovered an alternative but equally valid internal mechanism. Given that this assumption underlies the core analysis, a discussion of its implications and possible limitations would be needed.

- The analysis is limited to GAT, chosen for its attention mechanism, but this restricts the generality of the findings to other architectures, particularly MPNNs, which are often the most effective for neural algorithmic reasoning tasks.

- It is unclear how the proposed graph program language generalizes beyond the graph algorithms studied, especially since those are inherently parallel. Sequential algorithms such as Prim’s are not evaluated, so the broader applicability of the approach remains uncertain.

- The main result that there is no strong correlation between faithfulness and accuracy is interesting but somewhat limited in actionable insight. It would be valuable if the authors could discuss how these findings might guide the design of more faithful or generalizable NAR models.

- The external faithfulness measure based on comparing algorithmic traces is relatively straightforward. The observation that models perform well early in execution but degrade over time is somewhat expected, as errors naturally accumulate.

- The citation formatting does not follow ICLR style guidelines, which require author last names followed by the publication year. This can potentially impact the number of pages.

**Questions:**

- What are the different experimental settings in Table 6? More detailed explanations would help interpret the results.

- To what sizes did the test graphs generalize? It would be useful to analyze in-distribution versus out-of-distribution performance separately.

- How does the proposed approach differ from transformer-specific languages such as RASP? The conceptual similarities suggest the distinctions should be made clearer.

---

> ### Author Response · Authors · 2025-11-26
> **Thank you for your thorough review (Part 1)**
>
> Thank you for your kind and thorough review, and for highlighting the novelty and relevance of our work to the current state of the field.
>
> 1. We have added new sections justifying the use of attention patterns for comparison: Sections 4.1 and 4.5. For GATv2 specifically, the space of valid attention patterns is tightly constrained (because this is the only way to send information between nodes), justifying the use of a compiled version as ground-truth. While there may be some variability in attention patterns (e.g. for tie-breaking), a specific behavior is expected for each algorithm: BFS must send information to adjacent unexplored nodes, Bellman-Ford must select the minimum incoming neighbor, Bubble Sort must swap information between nodes, etc. . We address potential sources of variation, like tie breaking (CLRS defines a tie-breaking rule), and sharpness (see response to reviewer Kz1v). Section 4.5 specifically further validates the internal faithfulness metric, since we test 128 random initializations and none are close to our compiled solution (average pairwise similarity among learned models is 0.009 while best learned-to-compiled is 0.37). Our empirical analysis does not indicate any of them are within the distance expected from equally-valid attention patterns (e.g. sharpness).
>
> 2. It’s true that other NAR-literature architectures like MPNNs are of value and can outperform GATv2 (on some, but not all problems). While our metric is attention-specific, the broader compilation-to-trained comparison framework directly extends to other architectures, e.g. message patterns in MPNNs. We justify our choice of GATv2 in the Related Work section (lines 99-107). In particular, the similarity to the transformer architecture (but higher performance on CLRS) is the primary reason it was chosen for this paper. However, some of our main insights (e.g. underconvergence of hidden state modeling), have implications that affect other NAR architectures like TMPGNN.
>
> 3. We fully agree that sequential algorithms like Prim’s are a critical test case, though our main goal was to analyze high-performing algorithmically aligned algorithms, namely BFS and Bellman-Ford. Even analyzing just BFS creates interesting insights with implications for general NAR. We include Bellman-Ford analysis in the appendix. If time permits, we will include analysis from other algorithms, ideally bubble sort, binary search, and minimum.
>
> 4. We have included a new section (4.6) summarizing our actionable insights for NAR models based on our findings. In particular, the evidence of underconvergence leads to direct (and rather straightforward) ideas such as curriculum learning with variable learning rates between traces and predicted answers. Also, our experiments with architecture modification for GATv2 (which came from our compilation-based analysis) support trends in NAR (including several architectural choices for TMPGNN). We plan to update this section further in the next revision (Nov 30). Also see our response to reviewer NZTt, point 1.
>
> 5. We agree error accumulation is a natural explanation, but the sudden drop in performance is quite significant and coincides with the mechanistic gap, so we hypothesize it is more likely underconvergence in hidden-state modeling, which in turn affects overall mechanism learning (which depends on well-formed hidden states).

---

> > ### Author Response · Authors · 2025-11-26
> > **Thank you for your thorough review (Part 2)**
> >
> > (continued from above: questions onward)
> >
> > 1. See our dedicated Appendix section A with hyperparameter settings, the settings for these experiments are specifically in Tables 4, 5, and 6 (in the new revision: numbers have changed). We are currently running experiments which test underconvergence (e.g. curriculum learning), which will replace the old Table 6 in the final revision with curriculum learning experiments, and their mechanistic gap analysis (as well as experiments with the old settings). We plan to update these by November 30th.
> >
> > 2. We use the default CLRS settings, which generalize (usually, depends on algorithm) to length 32 (test) from lengths 4, 7, 11, 13, and 16 (train, regardless of algorithm). So the generalization is quite significant.
> >
> > 3. While our approach is conceptually similar, the compiled language is different. No previous analysis proposes using compiled programs for direct comparison like we do. The closest is ALTA, which analyzes the performance of transformers on Parity and SCAN, but doesn’t look at internal behavior like attention patterns.
> >
> > Thank you again for your thoughtful review and feedback. We hope these revisions address your main soundness concern, and demonstrate that the core contribution holds. If there is more we can clarify or update, please let us know.
> >
> > [1] Engelmayer, V., Georgiev, D. G., & Veličković, P. (2024, April). Parallel algorithms align with neural execution. In Learning on Graphs Conference (pp. 31-1). PMLR.
> >
> > [2] Greenlaw, R., Hoover, H. J., & Ruzzo, W. L. (1995). Limits to parallel computation: P-completeness theory. Oxford university press.

---

### Official Review · Reviewer_NZTt · 2025-10-27

**Soundness:** 2
**Presentation:** 3
**Contribution:** 2
**Rating:** 2
**Confidence:** 5

**Summary:**

In Neural Algorithmic Reasoning (NAR), neural networks (typically Graph Neural Networks) are trained to learn to execute algorithms. This paper studies whether NAR models actually learn execute algorithms by introducing two metrics to measure the faithfulness between what the model learns and the actual algorithm. This is done by creating a "ground truth model", i.e., a NAR model that perfectly performs an algorithm through Neural Compilation techniques. The trained model is then compared to the ground truth in terms of trace prediction accuracy (how well it follows the executing trace) and attention mechanism similarity. The results show that even when NAR models achieve very high downstream accuracy, they do not faithfully learn the real algorithms.

The main contribution of the paper consists in the definition of the two metrics, and the experimental study. Existing Neural Compilation techniques are used to obtain the "ground truth models".

**Strengths:**

- Using Neural Compilation to understand NAR is a novel and interesting idea. I think it could also lead ot potentially new strategies of training NAR models.
- The paper is well written and easy to follow.

**Weaknesses:**

- The analysis does not provide any real direction for improving NAR models
- The proposed internal metric is strictly tied to attention, so cannot be easily applied to other architectures
- Unless the Neural Compilation is unique, it is hard to make decisive claims. Could it be that a different compilation methods lead to better results?
- Some methodological choices need more justification (in particular the fact that the number of message-passing steps is set to 1, and that only a single attention head is used) as they differ from what is typically done in NAR

**Questions:**

- For the internal metric (eq. 25), would it make more sense to compare the rankings of the elements (in terms of attention scores) instead of the actual scores? Maybe a model can put a different distribution, but match the overall ranking of importance.
- I am not familiar with Neural Compilation, but it seems straightforward that the mapping from algorithm to parameters (line 111) is not unique (one way to look at it is simply to consider that permuting the neurons in a layer of an MLP leaves the output unchanged). This means that there can be a potentially very larger number of possible weight combinations that perfectly produce the given algorithm (i.e., a lot of different "compilations" leading to the same correct network). To understand the quality of the proposed metrics I think it is important to ask whether they would indicate perfect faithfulness to all possible weight combinations that perfectly "implement" the algorithm. I think this is true for the external metric, but is it also for the internal one?
Another aspect would be if two different "ground truth" networks would obtain the same scores, and I think this would not be the case for the internal metric.
I think the authors should add some discussions along these lines.
- It is true that attention offers a comparison that is easy to understand, but at the same time attention is only one component of the architecture, so it could be that the learned solutions might implement the algorithm via different internal mechanisms. Could the authors comment on this please?
- Related to above, it would be useful to quantify how much attention similarity should be expected between two learned models trained from different seeds?

---

> ### Author Response · Authors · 2025-11-26
> **Thank you for your careful review (Part 1)**
>
> Thank you for your careful review, and for recognizing the novelty of our paper. We appreciate your comment that you found the paper to be well-written.
>
> We’d clarify that the paper does not use existing neural compilation techniques — these are architecture specific, and this paper introduces neural compilation for GATv2 specifically (for example, compilers for ALTA target Universal Transformers, and don’t natively work for GATv2). While the empirical insights are our focus, this technique is also a contribution.
>
>
> 1. While our work does highlight specific improvements for NAR, we agree that they could be made more explicit, so we’ve added a new section 4.6: “Implications for NAR.” For GATv2 specifically, we highlighted architectural changes that reinforce existing literature, including ablation studies showing that our modification to edge-information passing is effective at improving Bellman-Ford performance. Furthermore, we show that the default training settings underconverge in terms of hidden state tracking and trace predictions (which also causes the internal mechanistic differences, in fact, they coincide over time, see Figures 2 and 4). For example, a remedy to this is variable learning rates between traces and answers, and a longer training process. We explicitly test this in our “grokking” experiments, but are expanding further and will include the results in our next revision (hopefully Nov 30). This underconvergence in terms of trace predictions and hidden state tracking is likely also a problem in other architectures, e.g. TMPGNN.
>
> 2. We argue our technique could be applied to other architectures of interest quite easily. In particular, we aim to target the popular Transformer architecture, which shares many similarities with GATv2 (see lines 99-107 for justification for why we chose GATv2). For architectures that do not have an attention layer (or other abstract mechanism), it is true that a different analysis would be required. However, the general approach of compiling in a solution as a ground truth is still a productive basis for such analysis, for any architecture. It’s worth noting that within NAR, even though some architectures outperform on some tasks, GATv2 is still competitive on other tasks, which is why we mainly analyze BFS and Bellman-Ford (which GATv2 does particularly well on).
>
> 3. Since we compare behavior in terms of attention, it isn’t critical for the compiled weights to be unique. In fact, the attention mechanism is tightly constrained and invariant to hidden state structure and exact parameters settings. In GATv2 specifically, attention is the only way to pass information between nodes, meaning that correct algorithm implementations must utilize it. Furthermore, CLRS defines a tie-breaking rule, further restricting the space of valid attention patterns. We’ve clarified this in the new sections of our paper, e.g. Section 4.1 and 4.3. Please also see our response to reviewer Kz1v. Also, we analyze across 128 random initializations, none of which come close to the compiled solution. See section 4.5, which shows that inter-solution differences are much smaller than the closest distance to the compiled solution.
>
> 4. Our methodological choices inherit from CLRS defaults, e.g. for GATv2, a single attention head and single message passing step is the default. Regarding attention heads, see our response to reviewer Kz1v. We’d argue the current CLRS-default settings offer an interesting and fair case to analyze, but see the value in generalizing our methods to other settings (and argue this would be relatively straightforward for future work).

---

> > ### Author Response · Authors · 2025-11-26
> > **Thank you for your careful review (part 2)**
> >
> > (continued from above: questions)
> >
> > 1. We appreciate your comment on attention sharpness (which ranking, your suggestion, would be invariant to), but maintain that the current choice of an L1 norm is appropriate, given that sharpness is a desirable property of a correct algorithm. Also, the difference between learned and compiled behavior is much larger than sharpness alone would account for. See Figure 3 for intuition supporting our choice, and Figures 2 and 5 for quantitative data, as well as section 4.5. Also see the response to reviewer Kz1v.
> >
> > 2. You’re correct that neurally compiled solutions are not unique in terms of weights. There are many symmetries of weights that produce the same behavior ([1]). However, this is not a flaw of the paper, because we intentionally create our metrics in terms of attention patterns! This captures abstract behavior with a much smaller space of possible correct implementations (e.g. tie-breaking behavior in Bellman-Ford or BFS). Also, we test a large number of random initializations (128), and find none of them come close to the compiled solution, indicating a systematic problem rather than an arbitrary ground truth. Also, the observed differences are too large to be accounted for by smaller algorithm details (e.g. tie breaking), and even the closest attention patterns (Figure 3) differ significantly. As to your question of how the metrics behave over the space of solutions: external faithfulness would be 100% for any correct solution. Internal faithfulness would also be 100% for any mechanistically faithful solution, with the exception of differences like tie-breaking. However, 1. The CLRS training data is created with a particular tie-breaking rule and 2. Our analysis does not indicate any learned algorithms that are correct except for a different tie breaking strategy, as they would be much closer to the compiled behavior and would at the very least have 100% external faithfulness.
> >
> > 3. While it could be valuable to characterize other components of the architecture, GATv2 (Graph Attention Networks) are arguably built around the attention mechanism as the primary component. Any information that passes between nodes is determined by this mechanism, and is required to correctly implement any of the algorithms in CLRS. We do include an experiment which adds edge-information passing to GATv2, and simplifies the decoding layers. This is done because we hypothesized that some important computation was happening in the decoding layers, because it couldn’t happen in the main model. Our results (Appendix) support this modification, which improves performance by about 4% and also improves faithfulness. For algorithms like BFS, the main model is capable enough to fully implement the algorithm without modifications (even at minimal parameter settings, e.g. <600 parameters), making it less likely that computation would need to be moved to the decoders.
> >
> > 4. This is an interesting question, and we’ve provided a dedicated analysis for it in Section 4.5, specifically in Figure 5. We find that inter-attention differences are much smaller than the closest distance to the compiled solution, and that the learned solutions form 18 different clusters of behaviors. Thank you for the suggestion!
> >
> > Thank you kindly for your thoughtful review. We hope that our revision alleviates your concerns and merits an increase in your score, especially given that you found our idea novel and interesting.
> >
> >
> > [1] Zhao, B., Walters, R., & Yu, R. (2025). Symmetry in Neural Network Parameter Spaces. arXiv preprint arXiv:2506.13018.

---

> > > ### Comment · Reviewer_NZTt · 2025-11-26
> > >
> > > Thank you very much for the detailed and thoughtful response.
> > > - Regarding the compilation of GATv2: I agree that this extension is useful. However, I want to note that my original point concerned the conceptual novelty: the methodology of compiling algorithms into neural parameters is well established, and the contribution here is a specific application rather than a departure from the known framework.
> > > - Regarding the uniqueness of compiled solutions and internal metrics: This remains my main concern. While you argue that the attention mechanism “tightly constrains” correct solutions, I am still not fully convinced that all valid implementations of the same algorithm must exhibit similar attention patterns. Furthermore the compiled solutions intentionally use "extreme" logits and biases to enforce sharp decisions. Learned models may implement comparable decision-making but using softer distributions.
> > > - Regarding the reliance on attention as the sole internal mechanism: I understand that that attention is the only way for information to pass between nodes in GATv2, but as also shown in your appendix B1, decoders can implement non-trivial computation, and that large amounts of logic may shift outside of the main attention mechanism
> > > - I appreciate your new experiments in Section 4.5. However, I think they still implicitly treat the compiled implementation as the sole ground truth behavior. I would encourage clarifying that your internal metric measures closeness to a particular mechanistic instantiation of the algorithm, not necessarily the only correct instantiation.
> > >
> > > I will raise my score to 4, but I believe the paper requires more work for publication.

---

### Official Review · Reviewer_Kz1v · 2025-10-30

**Soundness:** 2
**Presentation:** 4
**Contribution:** 3
**Rating:** 6
**Confidence:** 4

**Summary:**

This paper studies whether graph-based models for algorithmic reasoning actually implement the intended algorithms internally. The authors focus on GATv2 processors in the CLRS benchmark (BFS, Bellman–Ford, and DFS). They introduce a neural compilation procedure that analytically sets a GATv2’s parameters to execute a given graph algorithm. Faithfulness is then measured in two main ways, external hints, and internal (similarity between the learned attention distributions and those of the compiled reference). Empirically, then they find mechanistic gaps that models with near-perfect BFS accuracy still diverge from the compiled attention patterns, and trace predictions often degrade over time despite high final-answer accuracy.

**Strengths:**

I think internal faithfulness via attention-trace similarity really complements standard external trace metrics and predicted OOD accuracy, for the NAR community. I also really like the central empirical message of the paper that accuracy $\neq$ faithfulness and believe it is of importance for both NAR and interpretability community. I loved the writing style and clarity of the paper.

**Weaknesses:**

1. So, the internal metric compares learned attention to one compiled attention trace, but BFS and Bellman–Ford can admit many low-level realisations (e.g., different tie-breaking, temperature/scaling of logits, distributing computation into decoders). The paper works with algorithmic phase space and unique solutions, but does not formalize equivalence classes or prove that the chosen $\alpha*$ is the mechanism to match. The divergence might actually reflect an equally faithful but different mechanism. Please either justify uniqueness (up to known invariances) or report faithfulness against multiple compiled references (e.g., different tie-breakers) and/or invariance-aware distances.
2. I think external faithfulness mixes hidden-state quality with decoder power. Appendix A highlights that decoders can perform substantial computation, and the main text notes learned models are only good at predicting traces early on, possibly due to under-convergence or decoding (Fig. 3). Without fixing decoder capacity or probing states with a constrained linear probe (e.g 2210.13382 or 2408.14915), the external metric partly measures decoder design, not mechanism. I think using linear probes for core variables (visited, dist, $\pi$) would increase the soundness of the study. For example, If you probe $[visited, dist, \pi]$ directly from hidden states, how do external faithfulness curves change (Fig. 3/10)?

3. I suspect that there is a high possibility that a single-head constraint may mask faithfulness. All core results use one attention head because one head is sufficient, but there are evidences from various studies that emergent mechanistic modularity and expressivity might require multiple heads. Please report internal/external faithfulness with more heads (keeping capacity roughly fixed) to test whether additional heads close the gap.

**Questions:**

0. The questions above in the Weaknesses.
1. This is a personal concern. since the compiled BFS uses a pre-attention bias, doesn't it harm the length-generalization? The pre-attention bias simplifies compilation but ties size to problem size, breaking length generalization (Sec. 3.2; Table 8). So, my concern is that comparing a compiled model with the bias against learned models trained for length generalization is not apples-to-apples comparison. Can you elaborate of this?
2. Eq. (24) seems to miss a sum (it is ambiguous). please specify the sum/average over time and variables.
3. In Sec. 3.3, Eq. (18) for the predecessor looks wrong. for Bellman–Ford we expect ($\pi = \arg\min_j (d_j + w_{ij})$) and then copy ($x_j$) from that argmin. Please correct.
4. Upon more reading, I wonder whether you can quantify the "no correlation" claim, by providing a Pearson/Spearman coefficients between final accuracy and both faithfulness metrics for BFS and Bellman–Ford, and the sample size underlying.
5. Can you precisely define Eq. (25): which norm (I think it is l1), over which axes (nodes, edges, time), and what normalization? Did you try KL/JS/EMD? How sensitive are conclusions to temperature scaling of logits?
6. 2203.15544 and 2505.17190 (two important missing references on expressivity) argue that many DP-style algorithms live natively in the max-plus semiring and that softmax attention’s normalization blurs the underlying polyhedral decision structure. (the later) shows better OOD behavior when the attention kernel respects this tropical geometry. Given that your internal faithfulness compares softmax attention maps to a compiled softmax reference, could your observed mechanistic gap be partly a kernel/geometry mismatch rather than a learning failure? I think an analysis/discussion here would strengthen arguments of the study and create a deep dialogue in the NAR community.

---

> ### Author Response · Authors · 2025-11-26
> **Thank you for your careful review (Part 1)**
>
> Thank you for your careful, well-informed and constructive review of our paper.
>
>
> First of all, we appreciate your comments about our paper’s central message, its importance and novelty, as well as praise for the writing style and clarity of the paper.
>
> We’ve used your valuable feedback to revise the paper, adding new sections:
> 4.1: Defining Mechanistic Faithfulness
> 4.3: Internal Faithfulness
> 4.5: Validation of Faithfulness Metrics
>
> 1. We completely agree: in an ideal case, we would formalize equivalence classes. We would like to do this in a future work, since as much as we would like to do this, it is outside of the scope of the current paper. However, we believe the central empirical argument stands without them, since the compiled attention patterns represent the most mechanistically aligned patterns for GATv2 (which is tightly constrained). We argue for this in the new/rewritten sections of the paper (4.1-4.5).
> 1.a. Arguably, attention comparison already has (some) nice invariances, since it disregards exact weight settings and details of hidden state structure, effectively capturing abstract behavior. All that matters for our purposes is that information is passed between the correct nodes (which, for each algorithm, is required for a correct implementation). However, we agree with your points that more work in describing invariances would be valuable for validating our metrics. See sections 4.1 and 4.5
> 1.b. We experiment with 128 random initializations, and none of them come close to the attention behavior of the compiled solution, suggesting the gap is systematic rather than reliant on one algorithm implementation (section 4.3). Also, even the closest learned solution is significantly further than a compiled variant with different tie-breaking behavior. We validate this in section 4.5 and Figure 5, which analyzes differences in attention patterns between learned solutions, showing that inter-solution differences are much smaller than the closest distance to the compiled solution (0.009 vs 0.37).
> 1.c. Regarding tie-breaking: CLRS defines an arbitrary tie-breaking strategy in terms of node position, which we follow, but the difference expected from tie-breaking doesn’t account for the differences seen between learned and compiled solutions.
> 1.d. Regarding attention sharpness / temperature / ranking: This is a valid point, and we plan to include an explicit measure of how much sharpness matters, but would argue that the desired algorithm should have sharp attention behavior (e.g. 2505.17190, thanks for the fascinating reference).  Since an ideal algorithm has sharp behavior, we’d argue that the metric as-is actually has desirable properties, e.g. behavior that is close but not sharp has a better score than behavior that differs more significantly, which is what we see in the solution set.
> 1.e. Regarding computation in decoders: Within GATv2, there is an expected mechanism in terms of attention behavior, but architecture limitations (e.g. edge-information, which we analyze!) or training shortcuts may cause the model to move computation to unconventional places, e.g. skip connections or decoders. We consider these cases mechanistic failures, which is why we experiment with the edge-information variant of GATv2 (which also simplifies the decoders!).
>
> 2. We agree that external faithfulness partially measures the ability to decode hidden states, but argue this should not be a significant factor for a capable model. For many predicted elements, decoders are simple linear layers. Otherwise, we already include experiments (edge information experiment) that use a “simplified decoder” model, e.g. single linear layers. We also experimented with training only decoders when freezing the compiled internal model (in this case, the decoder easily learns a simple permutation). However, we agree that under-convergence is a factor (and argue it is a more likely cause than decoder power), which is why we include the “grokking” experiments (training 5x as long), (Table 9 in the current revision) . Since we realized under-convergence likely also explains internal mechanistic failures, we’ve actually revised the experiments to expand on this point (e.g. trying curriculum learning), and will include more detailed results on this before the end of the discussion period (hopefully by Nov 30).
>
> 3. Beyond being sufficient to capture the studied algorithms, single-head GATv2 is also the default setting of the CLRS benchmark. We agree that multi-head may learn more easily (e.g. under the Lottery Ticket or Scalar Bottleneck Hypotheses). However, given the high performance on BFS/Bellman-Ford under default settings, we think analyzing single-head attention is justified for the scope of this paper. Also, the high number of random initializations alleviates some of the concern with using a single attention head.

---

> > ### Author Response · Authors · 2025-11-26
> > **Thank you for your careful review (Part 2)**
> >
> > (continued from above, starting with questions section)
> >
> > 1. Pre-attention bias: Yes, this harms length generalization. Luckily, there are more sophisticated ways to achieve the same default-bias behavior without affecting length generalization, see [1]. Accordingly, pre-attention bias is something we intend to remove in a future version of this paper. In our specific comparisons, we separate experiments intended for length-generalization from experiments which require a pre-attention bias. In cases where a compiled and learned model are compared, both would have the same pre-attention bias setting, e.g. both with or both without. Generally pre-attention bias is only “required” (historically) for the compiled BFS experiment, and we try compiled Bellman-Ford experiments both with and without it.
> >
> > 2. Equations 24 and 25 have been updated. Thanks for mentioning this.
> >
> > 3. As-written the edge distances are added in equation 17, making equation 18 correct in terms of the updated distances. It is also correct to update both at once rather than sequentially, which is likely what you’re referring to.
> >
> > 4. Correlation coefficients were calculated to create Figures 1 and 2, and are now reported in the text as well (Table 1).
> >
> > 5. Equation 25 uses an L1 norm with summation over multiple axes: time, batches, and both node axes (in/out). While our conclusions aren’t sensitive to the sharpness of the attention coefficients (this would result in much smaller differences), we will include an analysis of sharpness in the next revision (Nov 30). See figure 3 for intuition on how attention tends to differ.
> >
> > 6. Thank you for providing these references. We agree that reference 2 in particular could be a factor, and we have cited it appropriately. At a glance, the mechanistic gap seems to be larger than kernel/geometry mismatch would account for, e.g. since hidden states do not seem to be fully tracked, which likely has downstream effects. So it seems more likely to be explained by underconvergence, e.g. if hidden states are not properly modeled, then the attention mechanism can’t even operate in terms of the correct information.
> >
> >
> > Thank you kindly for your encouragement and constructive suggestions. We have done our best to address your feedback, but plan to post another response by November 30th. Still, we hope that our current revisions address your concerns and merit raising your score. Please let us know if there are more points we can clarify.
> >
> >
> > [1] Ibarz, B., Kurin, V., Papamakarios, G., Nikiforou, K., Bennani, M., Csordás, R., ... & Veličković, P. (2022, December). A generalist neural algorithmic learner. In Learning on graphs conference (pp. 2-1). PMLR.

---

### Official Review · Reviewer_c4Jt · 2025-11-01

**Soundness:** 3
**Presentation:** 3
**Contribution:** 3
**Rating:** 6
**Confidence:** 2

**Summary:**

Authors introduce a neural compilation method for compiling algorithms into graph attention networks, and then utilize intermediate attention states of the compiled model as a reference for ideal behavior. They show that mechanistic gaps exist even for algorithms where NN is algorithmically aligned. On top of this authors propose several architecture modifications for target architecture (GATv2)

**Strengths:**

High accuracy on algorithmic reasoning tasks in distribution doesn't now imply that NN really learned algorithm and will have high OOD performance. Authors develop metrics (internal and external faithfulness) which could help better understand this behaviour.
On top of this they develop neural compilation method for compiling algorithms into graph attention networks  and improve GATv2

**Weaknesses:**

- limited number of algorithms considered
- internal attention mechanism similarity seems very interesting but it is not clear that 1-1 match should be expected due to potentially different representation or algorithm. would be good to explore this deeper

**Questions:**

Potentially there could be many different implementations for the same algorithm (and/or different representations). How do we know that the mechanistic gap found for BFS (internal faithfulness) is not due to this?

---

> ### Author Response · Authors · 2025-11-26
> **Thank you for your thoughtful review**
>
> Thank you for your thoughtful review.
>
> We appreciate your reinforcement of our core message, namely that accuracy doesn’t imply mechanistic faithfulness, and that our neural compilation technique in combination with our proposed metrics can be used to quantify and improve this behavior.
>
> As for uniqueness of algorithm implementations (and the validity of our internal faithfulness measure): Attention is an abstract behavior, so while multiple weight settings can represent the same algorithm, the space of attention mechanisms that correctly implement an algorithm is much smaller, which is why we’ve chosen attention patterns for our analysis. Also, attention behavior is invariant to details of hidden state structure (e.g. how cumulative distance is tracked in Bellman-Ford, or visited nodes in BFS). For GATv2 specifically, there are expected/required behaviors within the attention mechanism for each algorithm to be implemented correctly: BFS must send information to adjacent unexplored nodes, Bellman-Ford must select the minimum incoming neighbor, Bubble Sort must swap information between nodes, etc. Furthermore, in GATv2, information cannot be exchanged between nodes unless it is mediated by the attention mechanism.
>
> Furthermore, we analyzed 128 random initializations, finding that none come close to our compiled solution’s behavior in terms of attention (Figure 2). While there can be minor attention differences between correctly learned algorithms (e.g. tie breaking behavior), these do not account for the much larger mechanistic difference between the compiled and learned versions from our experiments (See Figure 3, the best matching BFS attention trace). Furthermore, CLRS specifies an arbitrary tie-breaking rule based on node position.
>
> Please see our new sections which address your feedback:
> 4.1: Defining Mechanistic Faithfulness
> 4.3: Internal Faithfulness
> 4.5: Validation of Faithfulness Metrics
>
> We selected BFS for the plots of the main paper body because it has such high OOD performance and (in theory) mechanistic alignment, yet shows a significant gap from expected behavior (both internally and externally). Our appendix includes a similar analysis for the Bellman-Ford algorithm. While we plan to include more algorithms (time-permitting: we are analyzing binary search, bubble sort, and minimum), we believe the core empirical message of the paper is already supported.
>
> Thank you kindly for your review and encouragement. Please let us know if there are further points on metrics that we can clarify, questions we can answer to help your confidence in the paper, or if there are other revisions you would like to see that would strengthen the paper.

---

### Author Response · Authors · 2025-11-26
**Overall Response**

Dear Reviewers,


Thank you for the insightful and constructive feedback, which has helped us identify key areas to strengthen. We are encouraged by the recognition of our work’s novelty and importance as well as praise of the paper’s writing and clarity by all four reviewers.

In response, we have created an updated manuscript, with new sections that address the main points of feedback. New content is added in blue.


Section 4.1, “Defining Mechanistic Faithfulness,” better clarifies and justifies the use of attention patterns to measure internal faithfulness. In particular, we argue that the attention mechanism in GATv2 is tightly constrained, and captures abstract behavior. We also clarify that attention patterns are invariant to hidden state structure or exact parameter settings. We address important points about uniqueness, e.g. tie-breaking, attention sharpness, and mechanisms outside of attention. We continue this in Section 4.5, “Validation of Faithfulness Metrics,” which includes a new analysis comparing attention differences within the set of learned solutions, and establishes mechanistic clusters that learned solutions fall into (Figure 5).

Also, we better describe and highlight the fact that our analysis uses a large number of 128 random initializations, which allows us to diagnose mechanistic failures at a systematic level, and reduces the choice of ground-truth as a factor in our analysis. We also highlight that external faithfulness is intended to validate the internal metric, and that the drop in both external and internal faithfulness occurs around the same time.

We have added another new section 4.6, which better comments on the insights gained from our analysis, namely internal underconvergence in hidden state modeling that could potentially be alleviated by longer training or curriculum learning. We plan to expand this section further, and will update the draft with results from experiments related to this.

We have supported Figure 1 with exact correlation coefficients and p values in Table 1 (both Pearson and Spearman), supporting one of our main claims that accuracy is not correlated with either internal or external faithfulness.

We clarified equations 24 and 25.

We have clarified training settings and methodological choices, and added discussion on how they enable a fair analysis.

We have included citations suggested by the reviewers.

We are continuing work and plan to upload another revision before the deadline, which will include more experiments targeting underconvergence (longer training, curriculum learning), experiments further solidifying the validity of the internal faithfulness metric (sharpness sensitivity), and (time-permitting) additional algorithms for the analysis (specifically: bubble sort, binary search, and minimum). However, we are confident in the current state of the paper and believe that the main empirical message (accuracy != faithfulness) is very well supported by the new revision.

We thank the ICLR reviewers again for their time and constructive comments. We believe these revisions significantly strengthen the paper and hope they address your concerns, potentially warranting an increase in scores and confidence. We look forward to any further discussion, and hope to provide more data and experiments before the discussion period is over. We are happy to provide further clarifications if needed.

---

### Meta-Review · Area_Chair_baHR · 2026-01-07

**Summary:**

This paper studies graph attention networks at the mechanistic level, trying to quantify, in the context of Neural Algorithmic Reasoning, to what extent the neural network has learned the correct mechanism to perform the task.


Overall, the main concern mentionned by all the reviewers was whether or not the measured mechanistic quantities (external trace prediction accuracy and internal attention similarity with the ground truth) were necessary.

I believe that this point is still outstanding and could be solved via either:
- a formal statement regarding the uniqueness of the mechanism to achieve the task.
- an additional experiment with specific instances for which it would be necessary to learn the desired internal mechanism to get high accuracy.

For these reason I considered this paper under the acceptance threshold.

**Reviewer Concerns:**

Overall, the main concern mentionned by all the reviewers was whether or not the measured mechanistic quantities (external trace prediction accuracy and internal attention similarity with the ground truth) were necessary.

In particular, it has been precisely articulated as follows by Reviewer Kz1v
> The paper works with algorithmic phase space and unique solutions, but does not formalise equivalence classes or prove that the chosen $\alpha^*$ is the mechanism to match. The divergence might actually reflect an equally faithful but different mechanism. Please either justify uniqueness (up to known invariances) or report faithfulness against multiple compiled references (e.g., different tie-breakers) and/or invariance-aware distances.

I believe it is the central problem to tackle theoretically in the work: showing formally that the described $\alpha^*$ is the **only** mechanism to match.

At least, the author could try to showcase specific instances for which failing to have a high attention similarity (or low external faithfulness) with the ground truth leads to inaccurate predictions. This would provide strong empirical evidence that learning $\alpha^*$ is necessary to perform the task correctly. At the moment, it is unclear whether learning such attention patterns is necessary, which weakens the conclusions can can be drawn from the experimental results.

**Reviewer Scores:**

I believe the reviewers would not have changed their score, which positions this paper under the acceptance threshold.

---

### Decision · Program_Chairs · 2026-01-26

Reject